# UV irradiation alters TFAM binding specificity and compaction of DNA

Dillon E King[1†], Emily E Beard[2†], Matthew J Satusky[3†], Alex George[1], Ian Ryde[1], Caitlin Johnson[2], Emma L Dolan[4,5], Yuning Zhang[6], Wei Zhu[7], Hunter Wilkins[2], Evan Corden[8], Susan K Murphy[5], Dorothy A Erie[2,9]*, Raluca Gordân[6,7,10,11]*, Joel Meyer[1]*

[1]Nicholas School of the Environment, Duke University, Durham, United States; [2]Department of Chemistry, University of North Carolina at Chapel Hill, Chapel Hill, United States; [3]Renaissance Computing Institute, University of North Carolina at Chapel Hill, Chapel Hill, United States; [4]Department of Pharmacology, Duke University, Durham, United States; [5]Department of Obstetrics and Gynecology, Duke University Medical Center, Durham, United States; [6]Department of Biostatistics and Bioinformatics, Duke University, Durham, United States; [7]Department of Molecular Genetics and Microbiology, Duke University, Durham, United States; [8]Program in Computational Biology and Bioinformatics, Duke University, Durham, United States; [9]Lineberger Comprehensive Cancer Center, University of North Carolina at Chapel Hill, Chapel Hill, United States; [10]Department of Computer Science, Duke University, Durham, United States; [11]Department of Genomics and Computational Biology, University of Massachusetts Chan Medical School, Worcester, United States

*For correspondence:
derie@unc.edu (DAE);
raluca.gordan@umassmed.edu
(RG);
joel.meyer@duke.edu (JM)

†These authors contributed
equally to this work

Reviewing Editor: Wolf-Dietrich
Heyer, University of California,
Davis, United States

## eLife Assessment

Mitochondrial DNA (mtDNA) exhibits a degree of resistance to mutagenesis under genotoxic stress, and this study on the mitochondrial Transcription Factor A (TFAM) presents **important** data concerning the possible mechanisms involved. The presented data are **solid**, technically rigorous, and consistent with established literature findings. The experiments are well-executed, providing **convincing** evidence on the change of TFAM-DNA interactions following UVC irradiation.

**Abstract** Mitochondria lack nucleotide excision repair; however, mitochondrial DNA (mtDNA) is resistant to mutation accumulation following DNA damage. These observations suggest additional damage sensing or protection mechanisms. Transcription Factor A, Mitochondrial (TFAM) compacts mtDNA into nucleoids and binds differentially to certain forms of DNA damage. As such, TFAM has emerged as a candidate for protecting mtDNA or sensing damage. To examine the possibilities that TFAM might protect DNA from damage or act as a damage sensing protein for irreparable forms of mtDNA damage, we used live-cell imaging and HeLa cell-based assays, atomic force microscopy (AFM), and high-throughput protein-DNA binding assays to characterize the binding properties of human TFAM to ultraviolet-C (UVC) irradiated DNA and the cellular consequences of UVC irradiation. Our cell data show increased TFAM mRNA after exposure and suggest an increase in mtDNA degradation without a loss in mitochondrial membrane potential that might trigger mitophagy. Our protein-DNA binding assays indicate a reduction in sequence specificity of TFAM following UVC irradiation and a redistribution of TFAM binding throughout the mitochondrial genome. Our AFM data show increased compaction of DNA by TFAM in the presence of damage. Despite the TFAM-mediated compaction of mtDNA in vitro, we do not observe any protective effect of increased TFAM protein on DNA damage formation in cells or in vitro. Increased TFAM protein did not alter

levels of mtDNA damage over time after UVC exposure in vivo, but knockdown of TFAM did alter mtDNA damage levels in HeLa cells both at baseline and after UVC exposure. Taken together, these studies indicate that UVC-induced DNA damage alters TFAM binding and promotes compaction by TFAM in vitro. We hypothesize that TFAM may act as a damage sensing protein in vivo, sequestering damaged genomes to prevent mutagenesis by facilitating removal or suppression of replication.

## Introduction

Mitochondria are complex organelles that play important roles in energy metabolism, cell signaling, immune response regulation, apoptosis, ion homeostasis, and other functions (*Picard and Shirihai, 2022*). Mitochondria contain their own circular double-stranded genome which is approximately 16.5 kilobases in size and encodes 13 proteins, all of which are essential subunits of the electron transport chain, along with two ribosomal RNAs and 22 transfer RNAs (*Gustafsson et al., 2016*). Mitochondrial DNA (mtDNA) is packaged into 'nucleoid' structures positioned at the inner mitochondrial membrane (*Lee and Han, 2017*; *Iborra et al., 2004*). The mitochondrial nucleoid is composed of over 50 nucleoid-associated proteins involved in maintaining mtDNA and regulating replication and expression (*Bogen-hagen, 2012*; *Spelbrink, 2010*; *Bogenhagen et al., 2003*; *Rebelo et al., 2011*). The most abundant protein present in the nucleoid is Transcription Factor A, Mitochondrial (TFAM). TFAM alone has been shown to be sufficient in vitro to condense and compact mtDNA (*Kukat et al., 2015*; *Kaufman et al., 2007*). Within a single mitochondrion, multiple copies of mtDNA exist (*Wachsmuth et al., 2016*; *Schon, 2000*) in differing conformations with regard to their nucleoid structure (*Isaac et al., 2024*). Notably, increased compaction is associated with decreased replication and gene expression activity (*Brüser et al., 2021*).

mtDNA is more susceptible than nuclear DNA to many damaging agents (*Zhao and andSumberaz, 2020*; *Meyer et al., 2013*), and it lacks some DNA repair pathways that protect the nuclear genome (*Zhao and andSumberaz, 2020*; *Meyer et al., 2013*; *Alexeyev et al., 2013*; *Gustafson et al., 2020*; *King and Copeland, 2025*; *Clayton et al., 1974*). Oxidative damage to mitochondrial genomes has been extensively studied because mitochondria are a major source of reactive oxygen species, but oxidative mtDNA damage is efficiently repaired by a robust base excision repair (BER) process. In contrast, the nucleotide excision repair (NER) pathway is absent in mitochondria (*Clayton et al., 1974*). The absence of NER is particularly salient, because it is the predominant DNA repair pathway that can repair damage caused by a wide range of very common environmental pollutants, including polycyclic aromatic hydrocarbons (PAHs), mycotoxins such as aflatoxin $B_1$, aromatic amines, ultraviolet (UV) light exposure, as well as some drugs such as cisplatin (*Gillet and Schärer, 2006*). The process of mitophagy, the selective degradation of mitochondria, contributes to the removal of irreparable mtDNA damage in human cells (*Bess et al., 2013*) and *Caenorhabditis elegans* (*Bess et al., 2012*; *Valenci et al., 2015*; *Pickrell et al., 2015*). Strikingly, though, despite often incurring high levels of NER-specific forms of DNA damage, the mitochondrial genome appears to be highly resistant to mutagenesis from these lesions. *Valente et al., 2016* did not detect mtDNA mutations in somatic cells of mice exposed to the PAH benzo[a]pyrene or the alkylating agent N-ethyl-N-nitrosourea, despite using exposures that caused nuclear DNA mutagenesis and high levels of mtDNA damage. We reported a similar result in the germline of the nematode *C. elegans* after 50 generations of continuous exposure to the DNA mutagens aflatoxin $B_1$ or cadmium, even in mitophagy-deficient mutants (*Leuthner et al., 2022*). The lack of mutagenesis in these mitophagy-deficient mutants suggests that an unknown cellular mechanism prevents mtDNA mutagenesis.

TFAM is a potential candidate for mutagenesis prevention, because it is known to compact the genome and has been suggested to be a damage sensing protein (*Chew and Zhao, 2021*). TFAM is a multifunctional protein that serves as both a transcription factor (*Campbell et al., 2012*; *Ngo et al., 2014*; *Kang et al., 2007*) and a core component of packaging mtDNA (*Kukat et al., 2015*; *Kaufman et al., 2007*; *Alam et al., 2003*; *Figure 1—figure supplement 1*). TFAM is an unusual transcription factor in that it binds specifically to promoters to initiate transcription but also exhibits high affinity binding throughout the mitochondrial genome, presumably to facilitate mtDNA packaging. TFAM is a high-mobility group (HMG) protein with two HMG binding domains that bind to DNA (*Ngo et al., 2014*; *Ngo et al., 2011*; *Rubio-Cosials et al., 2011*; *Choi and Garcia-Diaz, 2022*). Once bound to the DNA, TFAM can multimerize, which facilitates looping and ultimately compaction of the DNA. It

has previously been proposed that TFAM may serve as a DNA damage sensing protein for the mitochondrial genome (*Chew and Zhao, 2021*) as TFAM binds differentially to damaged vs. undamaged DNA in the context of oxidative damage (*Yoshida et al., 2002*; *Canugovi et al., 2010*), base loss (*Xu et al., 2023*), cisplatin adducts (*Yoshida et al., 2003*), bulged DNA (*Wong et al., 2009*), and alkylated lesions (*He et al., 2021*). TFAM also interacts with mtDNA repair processes, potentially playing a role in accelerating mtDNA degradation (*Xu et al., 2019*) and modulating base excision repair (*Canugovi et al., 2010*; *Urrutia et al., 2024*). While most of the forms of mtDNA damage for which TFAM interactions have been characterized can be repaired by mitochondrial base excision repair, interactions between TFAM and irreparable forms of mtDNA damage are less understood.

To further investigate mitochondrial responses to irreparable DNA damage, we used ultraviolet-C (UVC) irradiation. UVC irradiation is ideal for our experiments because it generates almost exclusively photodimers (*Friedberg and Walker, 2006*) that are irreparable in the mitochondria but repaired by NER in the nucleus. Using UVC nearly eliminates the occurrence of oxidative damage to DNA or other macromolecules caused by the reactive oxygen species generated by longer-wavelength UV radiation, which could confound interpretation. We characterized cellular responses to UVC exposure using live-cell imaging and gene expression in HeLa cells. We utilized atomic force microscopy (AFM) and protein-DNA binding chips coupled with biochemical assays to characterize human TFAM binding to UVC-induced lesions and structural changes to the compactional status of nucleoids following UVC-induced DNA damage. Additionally, we examined the dependence of UVC-induced DNA damage formation and DNA damage levels over time on TFAM protein levels, which has previously been shown to regulate compaction of mitochondrial nucleoids. Overall, our data show that cells respond to irreparable mtDNA damage by increasing mRNA of TFAM and other proteins involved in stimulating mtDNA turnover, in the absence of detectable loss of mitochondrial membrane potential or ATP. In vitro, we also characterize dramatic changes in TFAM binding to mtDNA sequences in the context of UVC-induced DNA damage and show that TFAM compacts UVC-damaged DNA more efficiently than undamaged DNA. In vivo, we found that increased TFAM does not protect against UVC-induced mtDNA damage nor alter levels of mtDNA damage over time after UVC exposure, but depletion of TFAM does alter mtDNA damage levels at baseline and after UVC exposure.

## Results

### Mitochondria respond to mtDNA damage from UVC exposure

We used multicolor live-cell imaging to investigate alterations to mitochondrial morphology and possible lysosomal trafficking of mtDNA following UVC exposures, after confirmation that these low doses of UVC are not associated with decreases in cell viability (*Figure 1—figure supplement 2*). We performed live-cell imaging on HeLa cells exposed to either 0 or 10 J/m$^2$ UVC following a 24 hr recovery time after the exposure, staining for mtDNA, lysosomes, mitochondria, and nuclei (*Figure 1A and B*) and quantifying the number of mtDNA spots, lysosome spots, area of mitochondria, and colocalization of lysosomes and mtDNA. Because we observe larger cells in the UVC-treated group (*Figure 1—figure supplement 3A*), we normalized our data to the area of each cell. Our data indicate a decrease in the area of mitochondria in UVC-treated cells (*Figure 1C*), but no significant differences in the mitochondrial morphology present in either treatment group, measured by the mean area of individual mitochondria or the mean perimeter (*Figure 1—figure supplement 3B and C*), although there is a modest decrease in mean eccentricity (*Figure 1—figure supplement 3D*). Notably, however, the cells exhibit an increase in the total number of lysosomes per cell (*Figure 1D*) and a decrease in the number of mtDNA spots (*Figure 1E*). Furthermore, a higher proportion of mtDNA spots colocalize with lysosomes in the UVC-treated cells versus the controls (*Figure 1F*, representative image in *Figure 1B*). Taken together, these results suggest increased mtDNA degradation.

We assessed levels of mtDNA damage immediately following exposure to UVC and after 24 and 48 hr of recovery time. Immediately after damage, we observe a dose-dependent increase in mtDNA lesions with UVC exposure (*Figure 2A*). After 24 hr, we observe a decrease in mtDNA damage levels, with significant levels of damage still present in cells exposed to 30 J/m$^2$ (*Figure 2A*). After 48 hr, damage is below the limit of detection (*Figure 2A*). The reduction in mtDNA damage could reflect an increase in mtDNA degradation removing the damage, an increase in mtDNA replication diluting the relative amount of damage present in the cells, or a combination of both.

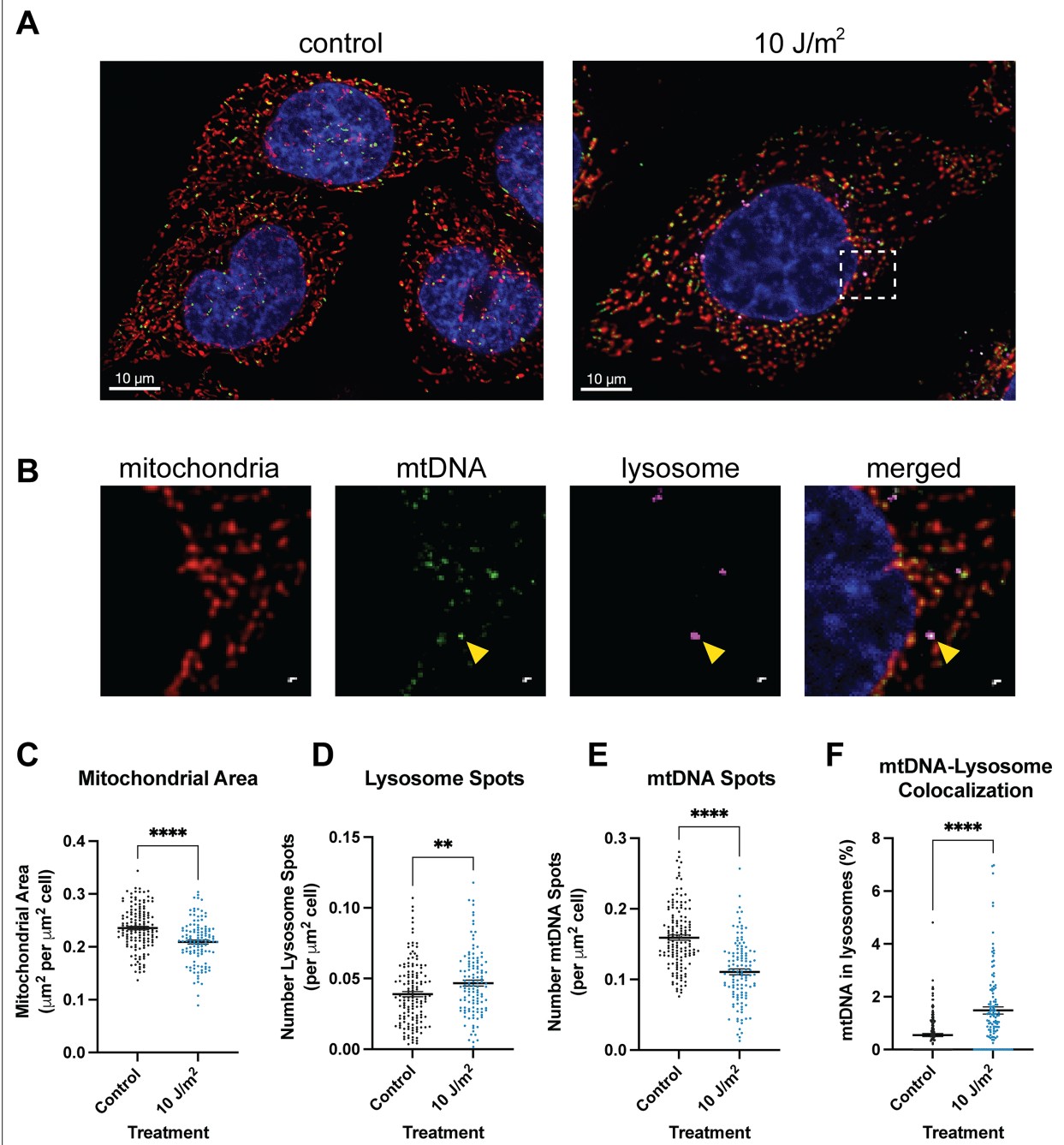

**Figure 1.** Live-cell imaging 24 hr following ultraviolet-C (UVC) treatment indicates increased mitochondrial DNA (mtDNA) degradation. (**A**) Representative images for control (left) and UVC-treated (right) cells. Merged channels include mitotracker (red, stains mitochondria), lysotracker (pink, stains lysosomes), picogreen (green, stains mtDNA), and nuclei (blue). (**B**) Inset outlined in the UVC-treated cells in panel A. Channels from left to right include mitochondria, mtDNA, lysosome, and the merged image. The yellow arrow indicates a mtDNA spot colocalized with lysotracker. For panels C-F, x-axes represent the treatment, and data was analyzed via two-tailed unpaired t-test. Data includes at least n=30 cells per treatment group per imaging session and three distinct imaging sessions. (**C**) Quantification of the total mitochondrial area per cell normalized to the size of the cell (p<0.0001). (**D**) Quantification of the number of lysosomes normalized to the size of the cell (*p*=0.0054). (**E**) Quantification of the number of mtDNA spots normalized to the size of the cell (*p*<0.0001). (**F**) Quantification of the proportion of mtDNA spots within lysosomes (*p*<0.0001).

The online version of this article includes the following figure supplement(s) for figure 1:

**Figure supplement 1.** Transcription Factor A, Mitochondrial (TFAM) roles in regulating the mitochondrial genome.

**Figure supplement 2.** Cell viability at 0-, 24-, and 48 hr following exposure to ultraviolet-C (UVC).

**Figure supplement 3.** Cell area and mitochondrial morphology analysis from live-cell imaging experiments.

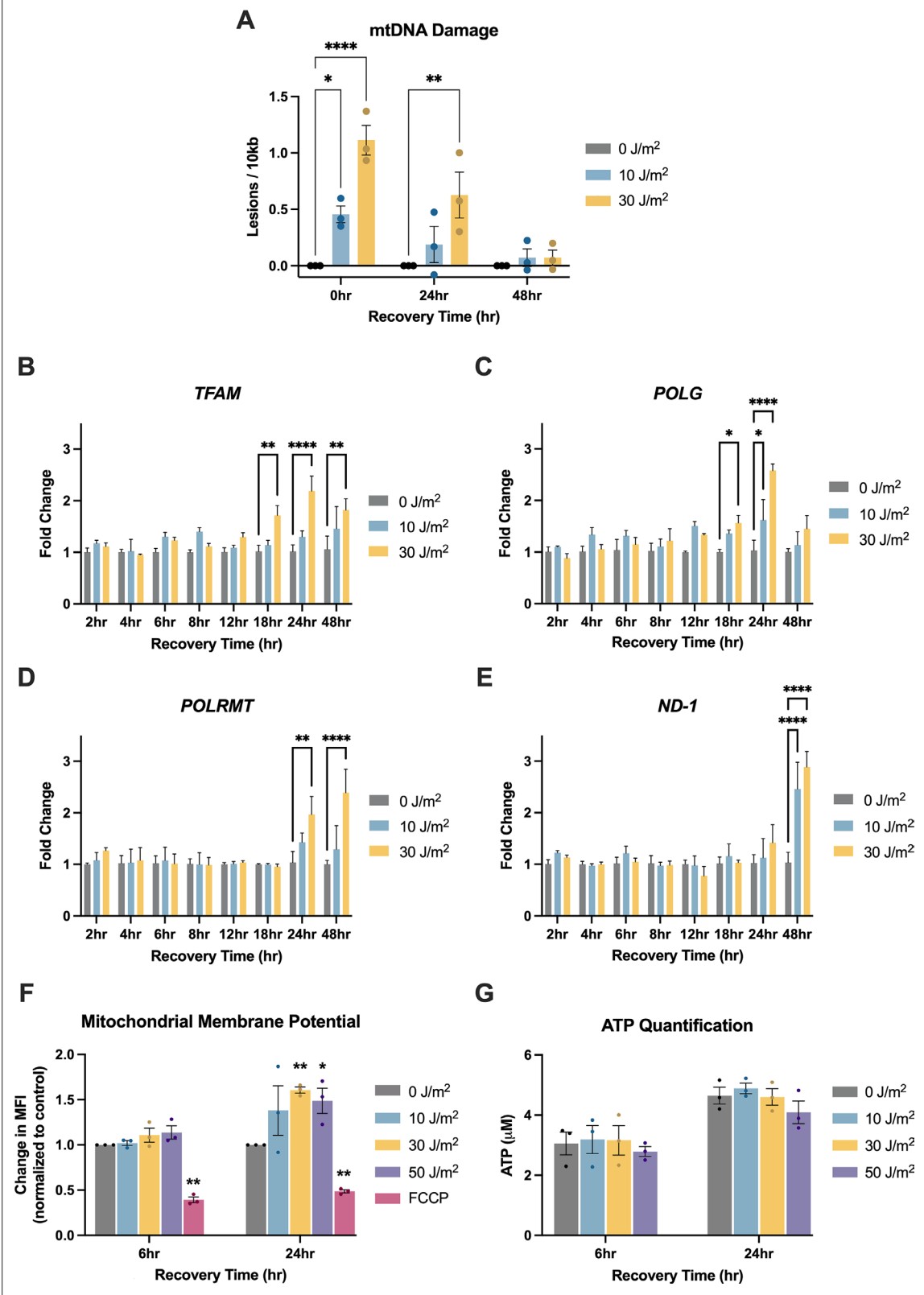

**Figure 2.** Ultraviolet-C (UVC) causes mitochondrial DNA (mtDNA) damage that decreases over time and is associated with upregulation of mtDNA replication genes, in the absence of apparent mitochondrial dysfunction. For panels A-F, x-axes represent recovery time, i.e., time following the UVC exposure. Doses of UVC used were 0, 10, and 30 J/m². (**A**) Mitochondrial DNA damage levels following UVC exposure. The y-axis represents the level of damage (lesions/10 kb). Data analyzed via two-way ANOVA with a Dunnett's post-hoc test for multiple comparisons (dose: $p<0.0001$, time: $p=0.0001$,

*Figure 2 continued on next page*

*Figure 2 continued*

interaction: *p*=0.0024). For panels B-E, y-axes represent fold change normalized to the control (0 J/m²) at each time point. All data was analyzed via two-way ANOVA with Dunnett's post-hoc test for multiple comparisons. (**B**) *TFAM* expression level assessed via qPCR following UVC exposure (dose: *p*=0.003, time: *p*<0.0001, interaction: *p*=0.01). (**C**) *POLG* expression level assessed via qPCR following UVC exposure (dose: *p*=<0.0001, time: *p*<0.0001, interaction: *p*=0.001). (**D**) *POLRMT* expression level assessed via qPCR following UVC exposure (Dose: *p*=0.002, time: *p*=0.008, interaction: *p*=0.05). (**E**) *ND-1* expression level assessed via qPCR following UVC exposure (Dose: *p*<0.0001, time: *p*=0.01, interaction: *p*=0.002). (**F**) Mitochondrial membrane potential assessed via flow cytometry following exposure to 0, 10, 30, or 50 J/m² UVC at 6 and 24 hr after exposure. The x-axis represents the exposure group and time point, and the y-axis represents the change in Median Fluorescent Intensities (MFI) of tetramethylrhodamine, methyl ester (TMRM) normalized to the control for each time point. Cells were also exposed to FCCP, a well-known chemical that causes mitochondrial depolarization, as a positive control. Data was analyzed via a two-way ANOVA (treatment: *p*=0.0008, time: *p*<0.0001, interaction: *p*=0.14). (**G**) Cellular ATP levels following exposure to 0, 10, 30, or 50 J/m² UVC at 6 and 24 hr after exposure. The x-axis represents the exposure group and time point, and the y-axis represents the ATP content (μM). Data was analyzed via a two-way ANOVA (treatment: *p*=0.38, time: *p*<0.001, interaction: *p*=0.94).

The online version of this article includes the following figure supplement(s) for figure 2:

**Figure supplement 1.** Ultraviolet-C (UVC) exposures alter gene expression of *PGC1α* and *NRF-1*.

To test for a transcriptional program supporting replication activity following UVC exposure, we assessed mRNA levels of key genes (*TFAM*, *PGC1a*, *NRF-1*, *POLRMT*, *POLG*, *ND-1*) associated with mtDNA replication and transcription. TFAM is responsible for initiating transcription, and it also plays a role in replication, because mtDNA transcription is coupled to mtDNA replication (*Falkenberg et al., 2024*). TFAM expression is regulated by NRF-1, which is regulated by PGC1α (*Scarpulla, 2011*). POLRMT is the mitochondrial RNA polymerase, which is necessary for transcription as well as replication as it generates primers to initiate mtDNA replication (*Falkenberg et al., 2024*). POLG is the catalytic subunit of the DNA polymerase responsible for DNA synthesis during mtDNA replication (*Graziewicz et al., 2006*). *ND-1* is a transcript of the mitochondrial genome that we utilized to assess mtDNA transcriptional activity.

We observe downregulation of *PGC1a* and *NRF-1* transcripts starting at 4 hr after exposure to UVC until 24 hr (*Figure 2—figure supplement 1A and B*), at which point *PGC1a* transcripts increase (*Figure 2—figure supplement 1A*), followed by elevation of *NRF-1* transcripts at 48 hr (*Figure 2—figure supplement 1B*). Interestingly, downregulation of *PGC1a* and *NRF-1* at the earlier time points is not associated with downregulation of *TFAM* (*Figure 2B*). We observe upregulation of *TFAM* (*Figure 2B*) and *POLG* (*Figure 2C*) beginning as early as 18 hr after the exposure to UVC. At 24 hr after exposure, we observe continued upregulation of *TFAM* (*Figure 2B*) and *POLG* (*Figure 2C*), as well as *POLRMT* (*Figure 2D*), but not *ND-1* (*Figure 2E*). These results suggest that within the first 24 hr after the UVC exposure, the cells' nuclear transcriptional program supports increased mtDNA replication, which could resynthesize non-damaged copies of the genome. After a 48 hr recovery period, *TFAM* (*Figure 2B*), *POLRMT* (*Figure 2D*), and *ND-1* are upregulated (*Figure 2E*), suggesting increased transcriptional activity. Taken together, the live-cell imaging results, quantification of mtDNA damage, and gene expression assays demonstrate a cellular response to increase mtDNA turnover that would support both removal and replacement of mtDNA.

A loss of mitochondrial membrane potential is an indirect mechanism that often allows for detection of mitochondrial dysfunction and induction of mitophagy; however, we did not detect a loss of membrane potential in cells exposed to UVC (*Figure 2F*). In fact, we observed a modest increase in membrane potential following 24 hr after the exposure (*Figure 2F*), which would not trigger mitophagy. We also tested whether the UVC exposure altered ATP levels, another potential indicator of mitochondrial dysfunction, but saw no differences (*Figure 2G*). We previously reported a lack of UVC-induced changes in mitochondrial reactive oxygen species levels in primary human fibroblasts (*Bess et al., 2013*). Taken together, these results suggest that the mtDNA turnover following damage does not result from indirect sensing of generalized mitochondrial dysfunction and further suggests that there may be a direct damage sensing mechanism.

## High-throughput in vitro data reveals that UVC-induced damage alters TFAM binding specificity across the mitochondrial genome

The preceding results suggest that cells might directly sense mtDNA damage. The observations that TFAM can compact DNA in vitro (*Kukat et al., 2015*; *Kaufman et al., 2007*) and exhibits differential binding to damaged DNA (*Chew and Zhao, 2021*) suggest that it could potentially serve as a damage

sensing protein. In addition, TFAM plays a role in maintaining mtDNA copy number and regulating both transcription and replication (*Campbell et al., 2012*; *Ngo et al., 2014*; *Kang et al., 2007*; *Ekstrand et al., 2004*). These functions may allow TFAM to either 'tag' genomes for degradation or sequester them from transcription and replication proteins. We hypothesized that TFAM would bind differentially to UVC-induced mtDNA damage, which would be required for such functions. However, in general, the specificity of TFAM binding across the mitochondrial genome is not well understood. There are conflicting reports on the specificity with which TFAM binds different regions of the mitochondrial genome, with some indicating specificity for promoter regions (*Blumberg et al., 2018*; *Malarkey et al., 2012*), and others reporting relatively uniform binding across the genome (*Wang et al., 2013*; *Miralles Fusté et al., 2014*). To address these gaps in knowledge and be able to compare TFAM binding to damaged vs. undamaged mtDNA, we leveraged an in vitro DNA-chip-based technology (*Berger et al., 2006*) to measure TFAM binding to tens of thousands of short DNA sites simultaneously.

We designed a custom library to cover the entirety of the human mitochondrial genome using a sliding 33-nucleotide window width with a 2-nucleotide shift for each consecutive probe (*Figure 3—figure supplement 1*). A subset of the chambers was irradiated with UVC to induce photolesions on the DNAs. We then incubated the chambers with either 30 nM or 300 nM TFAM (*Figure 3—figure supplement 1*). The concentrations of TFAM were selected based on preliminary experiments to ensure fluorescence intensity values were well within the range of detection. We were able to identify locations in the mitochondrial genome at which TFAM exhibits high occupancy and observed changes in sequence specificity of TFAM binding in the context of UVC-irradiated DNA.

To assess sequence specificity, we normalized TFAM binding levels by converting the fluorescence intensity values to z-scores using a control distribution based on 116 non-mitochondrial low affinity sequences previously identified in a TFAM universal protein-DNA binding array experiment (Methods, *Figure 3—figure supplement 2*). Our data reveal high-occupancy TFAM binding sites within the mitochondrial genome that are preferentially bound at the low concentration (30 nM) of TFAM, as indicated by the peaks in the z-scores plotted across the mitochondrial genome (*Figure 3*). At the high TFAM concentration (300 nM), peaks in binding occupancy occur at the same locations in the genome; however, their z-scores are lower due to the overall increase in TFAM binding to all sequences (*Figure 3—figure supplements 3 and 4*). We observe a strong correlation in z-scores between probes in the 30 nM and 300 nM datasets, further supporting the specificity of TFAM binding even at a high TFAM concentration (*Figure 3—figure supplement 4*). Previously, it has been suggested that the minimal binding motif for TFAM is two guanines separated by 10 random nucleotides, referred to as a $GN_{10}G$ motif (*Choi and Garcia-Diaz, 2022*) because all four TFAM promoter sequences contain this motif and crystal structures indicate interactions between TFAM and the guanine nucleotides in this motif (*Choi and Garcia-Diaz, 2022*; *Tan et al., 2022*). Our data, however, do not support this hypothesis. Specifically, the distributions of z-scores for sequences with and without the $GN_{10}G$ motif are strikingly similar, with slightly lower levels of TFAM occupancy indicated at sequences containing these motifs (*Figure 3—figure supplement 5*). Comparison of our data with previously published in vivo mitochondrial DNase I footprints (*Blumberg et al., 2018*) shows strong overlap between the two data sets (*Figure 3—figure supplement 6*), suggesting that our data may be representative of in vivo TFAM binding. Additional comparison of our data with recent in vitro TFAM footprinting performed on linear mtDNA using Fiber-seq (*Isaac et al., 2024*) indicates strong agreement between the two methods (*Figure 3—figure supplement 7*).

mtDNA contains four promoters: two on the heavy strand (HSP1 and HSP2) and two on the light strand (LSP1 and LSP2). Within promoter regions of the mitochondrial genome, we observe peaks in z-scores at both LSP1 and HSP1, but not LSP2 and HSP2 (*Figure 3*, *Figure 3—figure supplement 8*). However, the peak in HSP1 is located in the transcription start site rather than the TFAM binding site (*Figure 3—figure supplement 8*). Interestingly, the z-scores at these sites are lower than many of the high-occupancy sites identified throughout the genome (*Figure 3A*). Although perhaps surprising at first glance, previous work also indicates a lack of enrichment of TFAM at promoter sequences in vivo (*Wang et al., 2013*), and modest footprints of TFAM binding at promoter sites in vitro (*Tan et al., 2022*).

In contrast to the large peaks in the z-scores that we observe with non-irradiated DNA, experiments performed on irradiated DNA show more uniform binding patterns and reduction in size of distinct

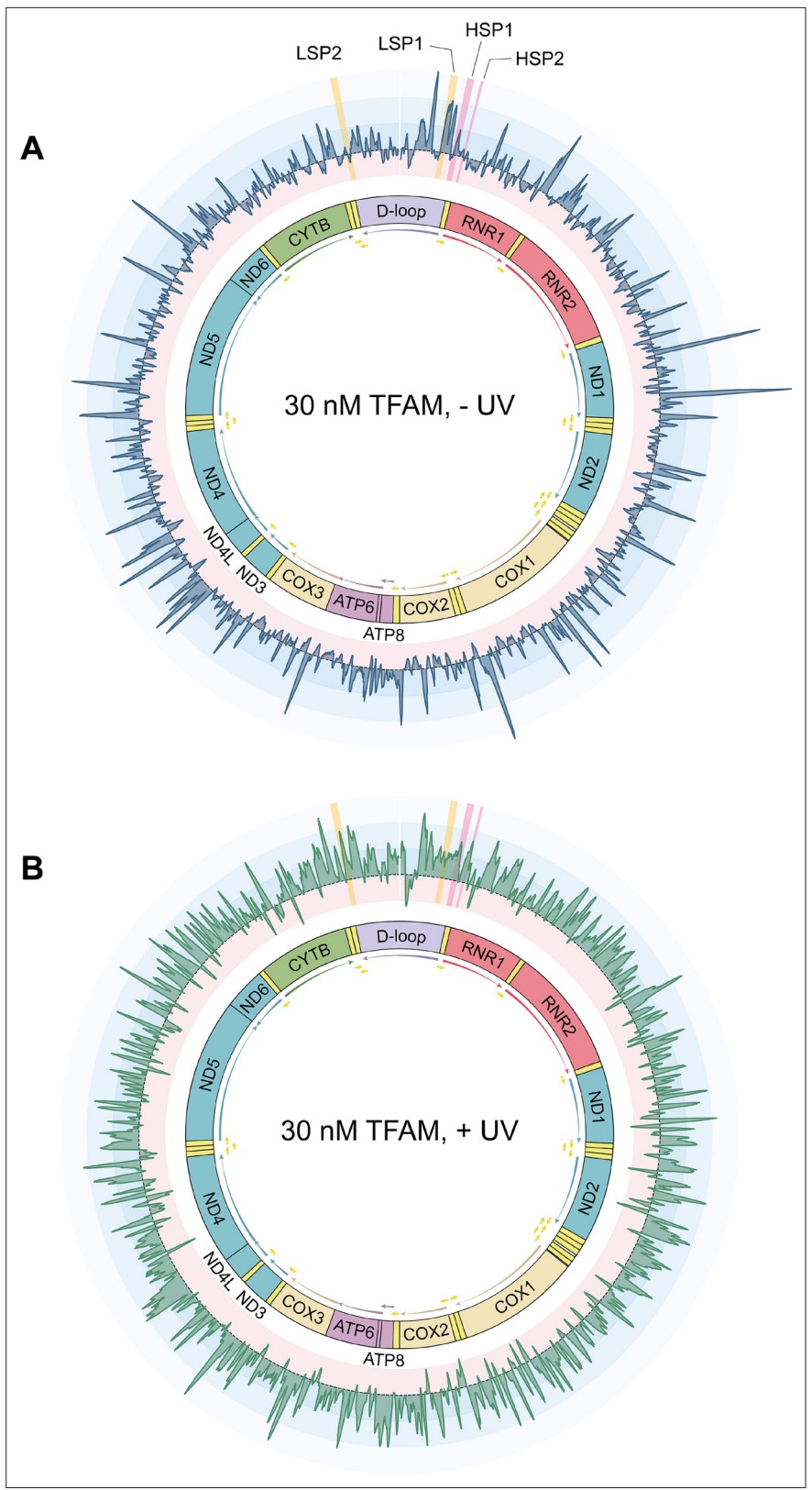

**Figure 3.** Transcription Factor A, Mitochondrial (TFAM) has specific binding across the mitochondrial genome and exhibits a reduction in specificity in the context of ultraviolet-C (UVC)-irradiated DNA. The median z-score is plotted to the coordinate of the middle nucleotide of the variable mitochondrial region of the sequence for the non-UVC-irradiated chamber containing 30 nM TFAM (**A**) and the chamber irradiated with 1500 J/m² UVC

*Figure 3 continued on next page*

*Figure 3 continued*

containing 30 nM TFAM (**B**). The gene map of the mitochondrial genome is shown in the center. Z-score variation is color-coded such that positive z-scores associated with high binding are in blue and progressively get lighter as the z-scores get higher. Negative z-scores associated with low binding are in red. Regions highlighted in yellow are the promoter sequences of the mitochondrial genome on the light strand (LSP1 and LSP2), while regions highlighted in pink are the promoter sequences on the heavy strand (HSP1 and HSP2). Plots of these regions can be found in *Figure 3—figure supplement 8*.

The online version of this article includes the following figure supplement(s) for figure 3:

**Figure supplement 1.** Custom DNA library with full coverage of human mitochondrial DNA (mtDNA) genome with a sliding window width of two nucleotides.

**Figure supplement 2.** Fluorescence intensity distribution of the bottom 116 non-mitochondrial sequences from a universal DNA-binding array used to calculate z-scores for each chamber.

**Figure supplement 3.** Experiments performed at 300 nM Transcription Factor A, Mitochondrial (TFAM) concentrations demonstrate that TFAM has specific binding across the mitochondrial genome and exhibits a reduction in specificity in the context of ultraviolet-C (UVC)-irradiated DNA.

**Figure supplement 4.** Correlation between z-scores obtained at 30 nM and 300 nM Transcription Factor A, Mitochondrial (TFAM).

**Figure supplement 5.** Across all experiments, $GN_{10}G$ motifs are not enriched in high Transcription Factor A, Mitochondrial (TFAM) occupancy groups.

**Figure supplement 6.** DNAse I footprints across the mitochondrial genome are enriched for sites with high in vitro binding signals for Transcription Factor A, Mitochondrial (TFAM).

**Figure supplement 7.** Comparison between fiber-seq high and low Transcription Factor A, Mitochondrial (TFAM) binding regions and our high-density TFAM-DNA binding array data.

**Figure supplement 8.** High-resolution view of Transcription Factor A, Mitochondrial (TFAM) binding to promoter sequences and shift in z-scores following ultraviolet-C (UVC)-irradiation.

**Figure supplement 9.** Ultraviolet-C (UVC)-irradiation is associated with a reduction in Transcription Factor A, Mitochondrial (TFAM) sequence specificity as weakest binders become tighter and tightest binders become weaker.

**Figure supplement 10.** Individual anisotropy plots for all sequences tested.

**Figure supplement 11.** Histogram plots of volumes for Transcription Factor A, Mitochondrial (TFAM) with and without DNA.

peaks in the z-scores throughout the mitochondrial genome (*Figure 3B*). These results suggest that the UVC damage reduces sequence specificity of TFAM. To examine this finding more systematically, we plotted the distribution of z-scores associated with the probes that are in the top 5% of z-scores and the bottom 5% of z-scores in the context of non-damaged DNA and UVC-irradiated DNA (*Figure 3—figure supplement 9A and B*). The z-scores of our top binders (*i.e.* probes for which TFAM exhibits high specificity) decrease after UVC irradiation (*Figure 3—figure supplement 9A*), whereas the z-scores of our weakest binding sequences (*i.e.* probes for which TFAM exhibits little to no preference) increase after UVC irradiation (*Figure 3—figure supplement 9B*). In addition, plotting the median z-score in the non-UVC-irradiated chamber against the change in z-score after UVC irradiation reveals a strong negative correlation, further supporting a reduction in sequence specificity upon UVC irradiation (*Figure 3—figure supplement 9C*).

## TFAM has high affinity for DNA and undergoes cooperative binding regardless of UVC-induced damage

The high-throughput binding assays reveal that TFAM exhibits DNA sequence preferences, but these preferences cannot be directly extrapolated to the binding affinity of TFAM to DNA, nor can they provide insight on the stoichiometry of binding. To investigate differences in TFAM binding affinity that may be associated with UVC-induced DNA damage, we performed fluorescence anisotropy and fit the binding curve to the Hill Equation (*Hill et al., 1910*) to determine the $K_D$ and the extent of cooperative binding (n) (Methods). Sequences were selected to cover a range of z-scores identified in our array-based binding assays, and anisotropy experiments were performed on both undamaged and UVC-irradiated oligonucleotides. All sequences analyzed exhibit $K_D$ values between 3 nM and

**Table 1.** Binding of Transcription Factor A, Mitochondrial (TFAM) to array sequences, measured using fluorescence anisotropy.

| Sequence | -UV | | | +UV | | |
|---|---|---|---|---|---|---|
| | Array z-score percentile | $K_D$ ±SEM (nM) | n±SEM | Array z-score percentile | $K_D$ ±SEM (nM) | n±SEM |
| ND4_473* | 0.48 | 4.95±0.03 | 5.06±0.09 | 28.28 | 5.36±0.14 | 4.76±0.11 |
| ND2_401 | 25.32 | 4.36±0.24 | 2.64±0.22 | 23.62 | 6.24±0.15 | 2.61±0.25 |
| ND3_92 | 25.68 | 4.53±0.13 | 2.63±0.22 | 10.23 | 5.54±0.14 | 2.19±0.06 |
| RNR2_619 | 25.71 | 4.89±0.33 | 2.46±0.23 | 51.19 | 8.13±0.39 | 2.33±0.10 |
| COX1_27 | 75.32 | 6.09±0.43 | 2.07±0.07 | 74.45 | 5.90±0.10 | 2.12±0.28 |
| ND1_288 | 75.16 | 3.41±0.03 | 3.04±0.19 | 35.79 | 6.12±0.22 | 1.90±0.12 |
| ND6_87 | 75.74 | 5.20±0.32 | 2.84±0.14 | 95.50 | 7.63±0.21 | 2.27±0.26 |
| ND1_450 | 90.25 | 4.33±0.17 | 2.40±0.43 | 70.40 | 6.36±0.51 | 2.20±0.13 |
| COX2_229 | 90.78 | 3.94±0.03 | 2.77±0.17 | 80.81 | 4.87±0.35 | 2.18±0.14 |
| TRNT_10 | 90.90 | 4.24±0.12 | 2.15±0.24 | 96.07 | 7.79±0.80 | 1.81±0.05 |
| ND1_353 40mer | 99.99 | 5.69±0.15 | 3.09±0.11 | 94.24 | 7.25±0.39 | 4.38±0.33 |
| ND1_353† | 99.99 | 7.56±0.34 | 3.85±0.49 | 94.24 | 7.66±0.88 | 2.94±0.40 |

*This sequence was used for AFM oligomerization studies to represent a low occupancy sequence in the array-based TFAM binding data, referred to as 'low occupancy sequence' in the text.

†This sequence was used for AFM oligomerization studies to represent a high occupancy sequence in the array-based TFAM binding data, referred to as 'high occupancy sequence' in the text.

9 nM and n values between 1 and 5, indicating multiple TFAM proteins binding to each DNA. Overall, we do not observe any large differences in $K_D$ or cooperativity values with or without UVC irradiation across any of the sequences analyzed (*Figure 3—figure supplement 10*, *Table 1*). This lack of correlation between the array-based binding data and the direct measurements of binding affinity contrasts with previous studies in which the occupancy of proteins at different DNA sequences, measured on high-density DNA arrays, directly correlates with the binding affinity (*Berger et al., 2006*; *Afek et al., 2020*; *Zhu et al., 2025*) and suggests that the differences in TFAM occupancy on the arrays may result from differences in extents of cooperative binding that are not discernable in the fluorescence anisotropy experiments.

To further examine the extent of oligomerization of TFAM on these DNA oligonucleotides and determine if there are any differences in TFAM binding on the 'high' and 'low' occupancy sequences (selected according to the array-based binding occupancy measurements), we used atomic force microscopy (AFM) to examine the oligomeric state of TFAM in the absence and presence of one of the high and one of the low occupancy sequences (*Table 1*). AFM is a powerful tool for determining the multimeric state of proteins, because the AFM volume correlates linearly with protein molecular weights (*Ratcliff and Erie, 2001*; *Yang et al., 2003*) and AFM can be used to identify cooperative binding on small DNA oligonucleotides (*Chelico et al., 2008*). Inspection of the volume distributions of TFAM in the absence of DNA reveals a single peak at ~35 nm³ with a tail to higher volumes, consistent with the dominant species being a monomer and the existence of a small amount of higher order species (Methods, *Figure 3—figure supplement 11A*). In the presence of the oligonucleotides, the volume distribution shifts to higher volumes that are consistent with dimers and higher-order species (*Figure 3—figure supplement 11B and C*). Notably, the 'high occupancy sequence' shows two distinct peaks at ~50 nm³ and 110 nm³, consistent with cooperative assembly of a dimer and tetramer of TFAM on this oligonucleotide. In contrast, the 'low occupancy sequence' exhibits a broad distribution with a peak at ~50 nm³ and a long tail to higher volumes, consistent with the cooperative formation of a dimer followed by weaker assembly of higher order species. This is consistent with previous SEC-MALS studies showing that TFAM oligomerization can differ with varying sequences (*Cuppari et al., 2019*). These observations in differences in binding properties may not be captured

in the fluorescence anisotropy measurement and, therefore, may account for the lack of correlation between the high-throughput array-based binding data and the binding affinities measured by fluorescence anisotropy.

## TFAM compacts UVC-damaged DNA more efficiently than undamaged DNA

The array-based TFAM binding data indicate that the relative occupancy of TFAM for sequences in the mitochondrial genome changes upon irradiation, with lower specificity associated with irradiation. Because one of the roles of TFAM is to compact the mitochondrial genome into nucleoid structures (*Kukat et al., 2015*; *Kaufman et al., 2007*), we examined the impact of UVC-induced DNA damage on nucleoid structure using AFM. For these experiments, we used circular plasmid DNA (pUC19) that was either untreated or exposed to 100 J/m$^2$ UVC, which introduces roughly three photolesions per plasmid (Methods). We incubated the damaged or undamaged DNA with TFAM for 2 min before depositing the sample on the mica surface. We used low concentrations of TFAM (15 nM or 30 nM) to prevent full compaction of the DNA and allow us to capture any differences in the extents of compaction.

*Figure 4A* shows images of pUC19 alone as well as undamaged and damaged pUC19 with 30 nM TFAM. The AFM images show that in the absence of TFAM, for both the undamaged pUC19 and the damaged pUC19, the plasmid is well spread on the surface (*Figure 4—figure supplement 1*). In contrast, TFAM promotes the formation of both punctate and disperse protein assemblies, as well as tracts of protein along the DNA, which often bridge two regions of DNA together (*Figure 4*, *Figure 4—figure supplement 2*). Notably, most DNAs show only a single cluster of proteins, with the majority of the rest of the DNA being unbound. In fact, at 15 nM TFAM, a significant percent of the DNAs have no protein bound, while other DNA molecules have punctate assemblies (*Figure 4—figure supplement 3*). These observations indicate that TFAM is cooperatively assembling on the DNA both in the presence and absence of irradiation, consistent with previous work indicating TFAM is highly cooperative (*Farge et al., 2012*).

To assess the extent of compaction, we measured the total volume of the DNA and its associated proteins on DNA (hereafter referred to as 'complexes') (Methods). In the absence of protein, the distribution of volumes shows a single peak centered at ~5000 nm$^3$ (*Figure 4B*). The addition of 15 nM and 30 nM TFAM results in the distributions broadening and shifting to higher volumes with increasing TFAM concentration in both the absence and presence of UVC irradiation. Notably, the volumes for the irradiated DNAs are greater than those without irradiation: the volumes for samples with irradiated DNA at 15 nM and 30 nM TFAM are ~12,600 nm$^3$ and ~12,700 nm$^3$, respectively, and the samples without irradiation at 15 nM and 30 nM TFAM are ~8250 nm$^3$ and ~7500 nm$^3$, respectively (*Figure 4B*). These results suggest that UVC irradiation increases the cooperative assembly of TFAM on pUC19. Inspection of the AFM images reveals that these differences in volume are associated with differences in compaction of the DNA. Specifically, addition of TFAM to UVC-irradiated DNA results in more compact and punctate structures relative to undamaged DNA, which shows proteins that appear to be more loosely associated with the DNA (*Figure 4A and C*, *Figure 4—figure supplement 3*).

To quantitatively assess the differences in compaction, we categorized each TFAM-pUC19 complex as: free DNA, dispersed (small clusters with no protein tracts on DNA), intermediate (small clusters with protein tracts), and punctate (tightly associated punctate clusters) (*Figure 4C*). At 15 nM TFAM, the majority of the DNAs (~55%) have no protein bound with very few punctate species with unirradiated DNA; whereas, the irradiated DNA shows significantly lower percentage of free DNA and an increase of punctate structures (*Figure 4D*, *Supplementary file 1*). At 30 nM TFAM, most of the DNAs show dispersed (~35%) and intermediate (~60%) species on unirradiated DNA and intermediate (~30%) and punctate (~60%) species on irradiated DNA. Taken together, these results indicate that UVC-induced DNA damage promotes compaction of DNA by TFAM (*Figure 4E*).

## Increased TFAM levels do not offer protection from UVC-induced mtDNA damage or alter accumulated DNA damage levels over time

Recent evidence shows that nucleoid compaction varies throughout cellular differentiation and between cell types (*Isaac et al., 2024*). It is well known that chromatin both protects from various

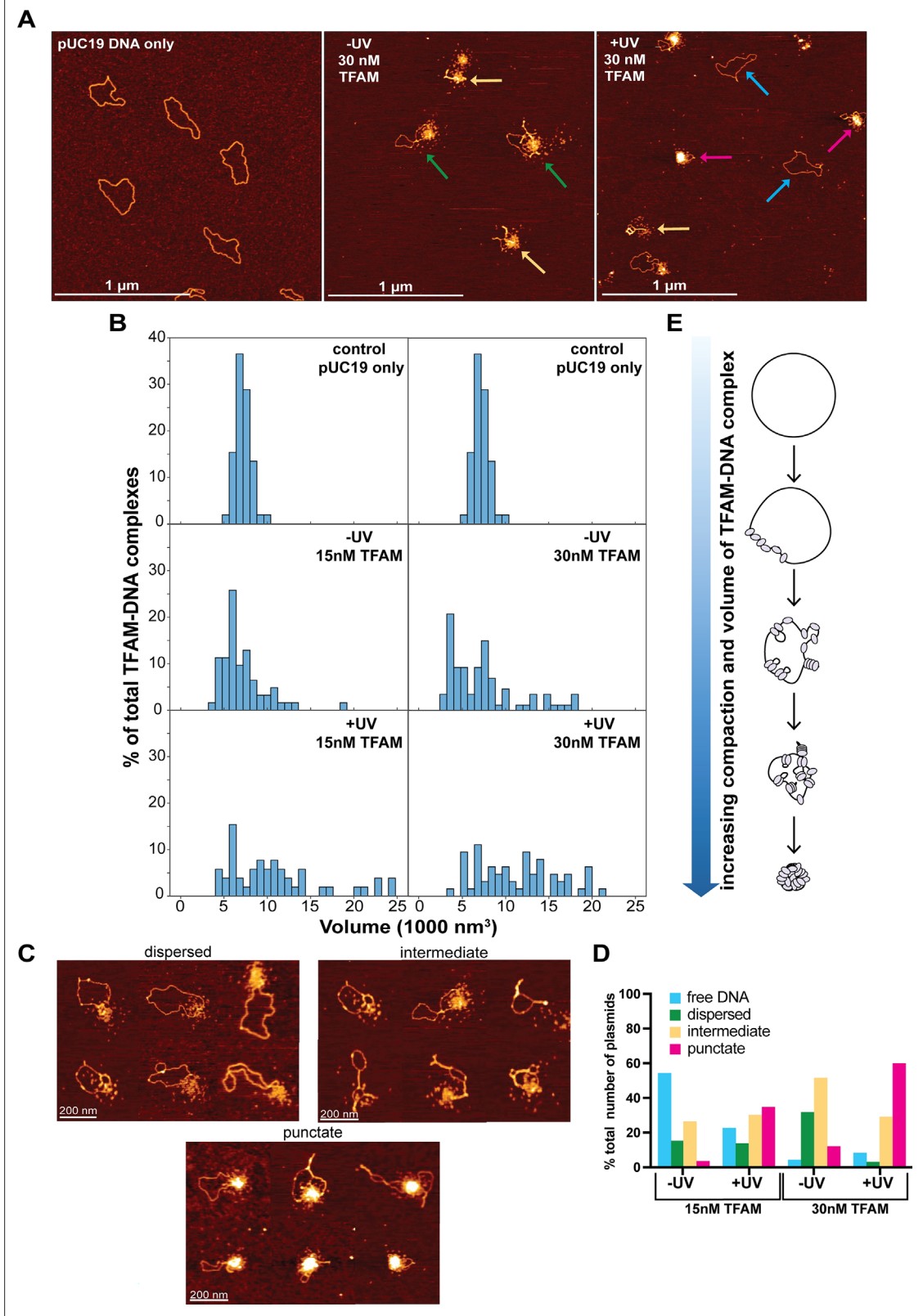

**Figure 4.** Atomic force microscopy of Transcription Factor A, Mitochondrial (TFAM)-DNA substrates indicates an increase in compaction associated with ultraviolet-C (UVC) exposure. (**A**) Atomic force microscopy (AFM) images of the pUC19 DNA only (control) and the 30 nM TFAM-DNA complexes in two different conditions: one with (+UV) or without (-UV) UVC irradiated DNA. Colored arrows on the AFM images represent the different TFAM-DNA complexes categorizations: dispersed (green), intermediate (yellow), punctate (red), and free DNA (blue). The white scale bar represents 1 μm.

*Figure 4 continued on next page*

*Figure 4 continued*

UV-irradiated plasmids were exposed to 100 J/m² UVC. (**B**) Histogram plot of the volumes distribution of plasmid DNA only (control) as well as the 15 nM and 30 nM TFAM-DNA complexes with (+UV) or without (-UV) UVC-damaged DNA. All axes in the histogram are scaled the same. The data for control pUC19 only was replicated for each TFAM concentration for clarity in comparisons. (**C**) Atomic force microscopy images of three different TFAM-DNA complex categorizations labeled as dispersed (small clusters with no protein tracts on DNA), intermediate (small clusters with protein tracts), and punctate (tightly associated punctate clusters). The white scale bar represents 200 nm. (**D**) A bar graph representing the percent total number of plasmids in the 15 nM TFAM concentration with (+UV) (N=66) or without (-UV) (N=136) UVC damaged DNA and the 30 nM TFAM concentration with (+UV) (N=65) or without (-UV) (N=91) UVC damaged DNA. Detailed counts of each classification can be found in *Supplementary file 1*. (**E**) Schematic of the TFAM-DNA binding and compaction mechanism.

The online version of this article includes the following figure supplement(s) for figure 4:

**Figure supplement 1.** Area quantification of undamaged and UV-irradiated pUC191341 plasmids using atomic force microscopy (AFM).

**Figure supplement 2.** 2D and 3D atomic force microscopy (AFM) images of Transcription Factor A, Mitochondrial (TFAM) tracts along DNA.

**Figure supplement 3.** Atomic force microscopy (AFM) images of 15 nM Transcription Factor A, Mitochondrial (TFAM) incubated with either damaged or undamaged DNA.

forms of DNA damage and regulates DNA repair activities (*Takata et al., 2013*; *Stadler and Richly, 2017*); however, little is known about the role of DNA compaction in the mitochondria. Although it has been proposed that the mitochondrial nucleoid may protect mtDNA from damage (*Brüser et al., 2021*), there is no direct evidence supporting this. Our observation that TFAM compacts damaged DNA suggests that mtDNA compaction may play a role in protecting the cell from damaged mtDNA. To examine these possibilities, we tested whether overexpressing TFAM, which has been shown to increase mtDNA compaction in cells (*Isaac et al., 2024*), would protect mtDNA from DNA damage or promote the removal of damaged mtDNA. To carry out these experiments, we used the same TFAM overexpression cell line generated by Isaac et al., which was previously used to show that increasing TFAM level increases the number of inaccessible (presumably compacted) nucleoids (*Isaac et al., 2024*).

Following a 48 hr pretreatment with doxycycline (schematic in *Figure 5A*), we observe a two-fold induction of TFAM protein, measured by western blot (*Figure 5B*, quantified in *Figure 5C*), as well as an increase in mRNA levels (*Figure 5D*). Elevated TFAM levels are sustained throughout the duration of the experiment (*Figure 5B and C*, *Figure 5D*). To test whether increased TFAM expression is associated with protection of mtDNA from UVC-induced DNA damage, we exposed these cells to UVC and quantified the amount of mtDNA damage they received. Across a range of UVC doses, there are no differences in the lesion frequency between the control and TFAM overexpressing cells (*Figure 5E*). Additionally, overexpression of TFAM does not influence the accumulated DNA damage levels over time, evidenced by the similar lesion frequencies observed 24 hr and 48 hr following the UVC exposure (*Figure 5F and G*), nor does overexpression of TFAM protect the cells from a loss of cell viability following exposure to UVC (*Figure 5—figure supplement 1*).

To test if the level of compaction of the mitochondrial nucleoid is protective against mtDNA damage across a wider range of TFAM concentrations, we generated in vitro reconstituted nucleoids with full-length human mtDNA and varying amounts of TFAM. We generated a series of dose-response curves for UVC to determine if elevated TFAM levels are associated with protection from UVC-induced DNA damage. The conformational status of the mtDNA nucleoids was confirmed using AFM (*Figure 5H*). We observe that increased nucleoid compaction, driven by elevated TFAM concentrations, is not sufficient to protect mtDNA from UVC-induced lesions (*Figure 5I*). Taken together, these results indicate that the degree of TFAM-mediated compaction of mtDNA does not affect the amount of DNA damage from UVC, and that a roughly twofold increase in TFAM protein does not alter the accumulated mtDNA damage levels over time. Finally, we tested whether knockdown of TFAM would alter mtDNA damage levels in HeLa cells. We found that TFAM knockdown progressively increased mtDNA damage over time in the absence of UVC exposure, as previously reported (*Canugovi et al., 2010*), and resulted in a persistence of mtDNA damage over time after UVC exposure (*Figure 5J*), likely in part due to the reduced mtDNA copy number and replication capacity following the knockdown of TFAM (*Figure 5K*).

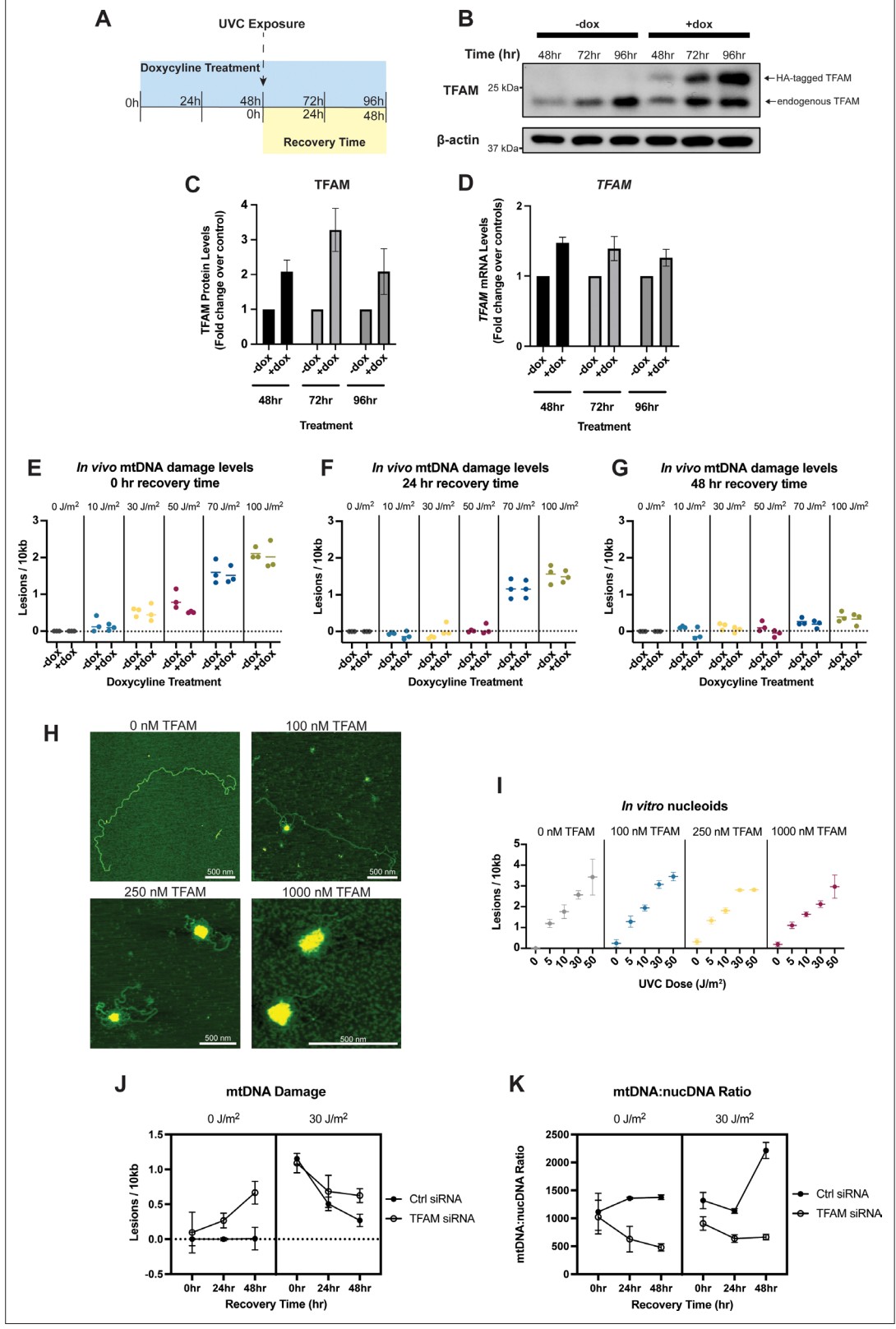

**Figure 5.** Increased Transcription Factor A, Mitochondrial (TFAM) levels in vitro or in vivo do not protect mitochondrial DNA (mtDNA) from ultraviolet-C (UVC)-induced DNA damage or alter damage levels over time. (**A**) Exposure paradigm for TFAM overexpression experiments. Cells were exposed to doxycycline for 48 hr prior to UVC exposure to ensure upregulation of TFAM at the time of exposure. Protein was quantified in control cells

*Figure 5 continued on next page*

*Figure 5 continued*

only to ensure TFAM upregulation. (**B**) Representative western blot of TFAM-tetON cell lysates following 48, 72, and 96 hr of doxycycline treatment to confirm TFAM upregulation. TFAM-tetON cells contain an HA tag that when expressed, results in a second band. (**C**) Pixel quantifications of n=3 western blots shown in panel (**B**) at each time point. TFAM protein levels were normalized to β-actin and then normalized to the non-doxycycline-treated controls. The x-axis represents the doxycycline treatment and y-axis represents the fold change relative to the non-doxycycline-treated cells. Data was analyzed via two-way ANOVA (doxycycline treatment $p=0.0006$, time $p=0.25$, interaction $p=0.25$). (**D**) *TFAM* mRNA quantification following doxycycline treatment. The x-axis represents the doxycycline treatment, and the y-axis represents the fold change relative to the non-doxycycline-treated cells. Data was analyzed via two-way ANOVA (doxycycline treatment $p=0.0003$, time $p=0.52$, interaction $p=0.52$). (**E**) Lesion frequency following UVC exposure in TFAM-tetON cells immediately after the exposure (0 hr recovery time). The x-axis represents the doxycycline treatment across a range of UVC doses and y-axis represents the lesion frequency (lesions per 10 kb). Data was analyzed via two-way ANOVA (UVC dose: $p<0.0001$, doxycycline treatment $p=0.15$, interaction: $p=0.77$). (**F**) Lesion frequency following UVC exposure in TFAM-tetON cells 24 hr after the exposure (24 hr recovery time). The x-axis represents the doxycycline treatment across a range of UVC doses and y-axis represents the lesion frequency (lesions per 10 kb). Data was analyzed via two-way ANOVA (UVC dose: $p<0.0001$, doxycycline treatment $p=0.71$, interaction: $p=0.65$). (**G**) Lesion frequency following UVC exposure in TFAM-tetON cells 48 hr after the exposure (48 hr recovery time). The x-axis represents the doxycycline treatment across a range of UVC doses and y-axis represents the lesion frequency (lesions per 10 kb). Data was analyzed via two-way ANOVA (UVC dose: $p<0.0001$, doxycycline treatment $p=0.01$, interaction: $p=0.61$). (**H**) Representative atomic force microscopy (AFM) images of in vitro nucleoids generated using purified TFAM and PCR amplified human mtDNA at 0, 100, 250, and 1000 nM TFAM. All scale bars represent 500 nm. (**I**) Lesion frequency following UVC exposure in in vitro nucleoids. The x-axis represents the UVC dose and y-axis represents the lesion frequency (lesions per 10 kb). Data was analyzed via two-way ANOVA (UVC dose: $p<0.0001$, TFAM concentration $p=0.20$, interaction: $p=0.86$). (**J**) Lesion frequency following UVC exposure in cells with and without a TFAM knockdown. The x-axis represents the recovery time, i.e., time following exposure to UVC, and the y-axis represents the level of mtDNA damage (lesions per 10 kb). Data was analyzed via a three-way ANOVA (UVC dose: $p<0.0001$, recovery time: $p=0.002$, siRNA treatment: $p<0.0001$, recovery time*UVC dose: $p<0.0001$, recovery time*siRNA treatment: $p=0.001$, UVC dose*siRNA treatment: $p=0.08$, recovery time*UVC dose*siRNA treatment: $p=0.66$). (**K**) Ratio of mtDNA copy number to nuclear DNA copy number following UVC exposure in cells with and without a TFAM knockdown. The x-axis represents the recovery time, i.e., time following exposure to UVC, and the y-axis represents the level of mtDNA damage (lesions per 10 kb). Data was analyzed via a three-way ANOVA (UVC dose: $p=0.01$, recovery time: $p<0.01$, siRNA treatment: $p<0.0001$, recovery time*UVC dose: $p<0.001$, recovery time*siRNA treatment: $p<0.0001$, UVC dose*siRNA treatment: $p=0.04$, recovery time*UVC dose*siRNA treatment: $p=0.01$).

The online version of this article includes the following source data and figure supplement(s) for figure 5:

**Source data 1.** Original files for western blot analysis displayed in *Figure 5B*.

**Source data 2.** PDF file containing original western blot analysis displayed in *Figure 5B*, indicating relevant bands, treatment groups, and time points.

**Figure supplement 1.** Transcription Factor A, Mitochondrial (TFAM) overexpression does not protect cells from loss of cell viability from ultraviolet-C (UVC) exposures.

# Discussion

The mechanisms by which cells handle mtDNA damage remain poorly understood. In particular, the absence of NER in mitochondria begs the question of how bulky lesions and photodimers in mtDNA are removed. TFAM is a multifunctional protein that regulates both transcription and replication of mtDNA, and it compacts the genome into nucleoids. Previously, other groups have hypothesized that TFAM may protect the DNA from DNA damage (*Brüser et al., 2021*) and serve as a damage sensing protein (*Chew and Zhao, 2021*). In this work, we investigate cellular responses to low levels of mtDNA damage caused by UVC irradiation as well as the effect of UVC damage on TFAM-DNA interactions. The results from this study indicate that although TFAM does not protect against UVC-induced DNA damage, it does alter mtDNA compaction in vitro, alters mtDNA damage levels with and without UVC exposures in vivo, and may function as a photolesion DNA damage sensing protein. This work represents the first step towards understanding how TFAM may interact with irreparable NER-specific substrates to prevent mutagenesis in the mitochondrial genome.

## Mitochondria respond to UVC-induced mtDNA damage in the absence of apparent mitochondrial dysfunction

Exposure to UVC that induces mtDNA damage results in a host of responses, including removal of mtDNA and an increase in gene expression of key mtDNA replication genes, presumably to replenish healthy copies of mtDNA (*Figures 1 and 2*). The removal of mtDNA begins within the first 24 hr after damage exposure, indicated by an increase in mtDNA present within lysosomes as well as a decrease in the number of mtDNA spots present in live-cell imaging experiments (*Figure 1*). This removal of mtDNA occurs alongside upregulation of *TFAM*, *POLG*, and *POLRMT*, all of which are involved in mtDNA replication (*Figure 2*). The observed decrease in mtDNA damage levels over time after UVC exposures (*Figure 2A*) could be a result of both the removal of mtDNA and an increase in the synthesis of non-damaged mtDNA molecules; additional work will be required to distinguish these possibilities. 48 hr after the exposure to UVC, cells upregulate *TFAM* and *POLRMT* as well as transcriptional activity on mtDNA (measured by upregulation of *ND-1*) (*Figure 2*), likely to replenish mtRNA and protein levels. These damage-induced changes to mtDNA transcription and the transcription of nuclear-encoded genes involved with mtDNA replication (*TFAM*, *POLG*, *POLRMT*) occurred in the absence of apparent mitochondrial dysfunction, indicated by no detectable changes to mitochondrial membrane potential or ATP levels (*Figure 2F and G*). Previously, we have also shown that UVC exposures do not result in a detectable increase in mitochondrial reactive oxygen species (mtROS) (*Bess et al., 2013*), though it should be noted that this study used a different type of cell and included a thymidine block. While other possible manifestations of mitochondrial dysfunction after UVC exposure cannot be ruled out, the current absence of any detectable indications of mitochondrial dysfunction following UVC irradiation suggests the possibility that cells might directly detect UVC-induced mtDNA damage.

## TFAM exhibits preferential binding to a large number of sequences in the mitochondrial genome

Examination of the array-based binding data reveals that in the absence of damage, TFAM shows a wide range of z-scores across the mitochondrial genomic sequences (*Figure 3*), indicating preference of certain sequences. These data are in agreement with previously published DNase footprinting data (*Blumberg et al., 2018*; *Figure 3—figure supplement 6*) and the recently published Fiber-seq footprinting technique used to map TFAM binding patterns to linear mtDNA (*Isaac et al., 2024*; *Figure 3—figure supplement 7*). In contrast, previously published ChIP-seq data for TFAM does not show specific enrichment, i.e., no significant peaks (*Wang et al., 2013*; *Lee et al., 2022*). However, given the heterogeneity of compaction of mitochondrial genomes in cells and the fact that many genomes in each cell are fully coated with TFAM (*Isaac et al., 2024*; *Brüser et al., 2021*), it is not surprising that ChIP assays do not result in strong TFAM enrichment patterns. Interestingly, the sequences in our data with high z-scores are not associated with the $GN_{10}G$ motif, which was previously suggested as a consensus sequence for TFAM (*Figure 3—figure supplement 4*; *Choi and Garcia-Diaz, 2022*). However, this is not surprising given that the higher affinity of TFAM for this motif in the original publication (*Choi and Garcia-Diaz, 2022*) was only observable at high salt concentrations but not at physiologic salt concentrations like those utilized in our study. This is consistent with Fiber-seq analysis of TFAM binding, which also failed to observe significant enrichment at $GN_{10}G$ motifs in the mitochondrial genome (*Isaac et al., 2024*). In addition, we found that the z-scores at promoter sites were modest compared to other regions in the genome (*Figure 3*, *Figure 3—figure supplement 8*). These findings indicate that, unlike most transcription factors, TFAM by itself shows low specificity for its promoters, with the exception of LSP1, and suggest that TFAM may require interactions with additional proteins present in the nucleoid to promote transcription. Indeed, both earlier work from *Gaspari et al., 2004* and recent work from Tan et al. (see Figure S4a in *Tan et al., 2022*) indicates that POLRMT and TFB2M are also needed, in addition to TFAM, to produce a notable enrichment of TFAM at the promoters. Although we do not see strong binding signals at the promoter sites other than LSP1, the preferential binding of TFAM to sequences across the genome may be involved with facilitating compaction of the genome, which could in turn regulate transcription and replication. Specifically, the extent of the compaction of the mitochondrial genome is inversely proportional to the transcription and replication activity (*Brüser et al., 2021*), similar to nucleosomes.

While it is clear from our array data and other studies (*Isaac et al., 2024*; *Blumberg et al., 2018*) that TFAM exhibits sequence specificity, no specific binding motifs have been identified. The sequences we determined to have the highest TFAM binding levels on our array do not overlap with known regulatory or structural elements within the mitochondrial genome, such as origins of replication, promoter sites, or secondary structures that may form on the arrays. There is, however, one exception to this in the D-loop where we observe a peak in TFAM binding level upstream of LSP1 at the conserved sequence block 2 (CSB2) (*Figure 3A*). Interestingly, there is a discrepancy between our TFAM binding array results and our determined $K_D$ values for these same sequences via fluorescence anisotropy (*Table 1*) in that sequences for which we observe stronger TFAM signal on the arrays do not have tighter binding affinity. This suggests to us that the differences in TFAM binding levels that we observe on the arrays may be driven by another feature, such as differences in TFAM cooperativity. Indeed, using AFM, we determined that for a sequence for which we observe higher binding levels on the array, TFAM undergoes increased oligomerization compared to a sequence for which we observed lower TFAM binding levels on the array (*Figure 3—figure supplement 11*). It remains unknown what about these specific sequences drives differences in TFAM cooperativity. Given that TFAM is known to bend DNA in order to facilitate compaction (*Farge and andFalkenberg, 2019*) and has been proposed to induce small ssDNA bubbles (*Farge et al., 2012*), one possibility is that differences in either DNA flexibility or local melting abilities could be influencing TFAM's sequence specificity, but this remains to be tested.

## TFAM does not protect DNA from UVC-induced damage but more readily compacts irradiated DNA

Previous studies have indicated that TFAM promotes DNA compaction (*Kukat et al., 2015*; *Kaufman et al., 2007*). Examination of complexes of TFAM bound to pUC19 and linear full-length human mtDNA using AFM shows that TFAM promotes compaction of both DNAs, with the extent of compaction dependent on TFAM concentration (*Figure 4*, *Figure 4—figure supplement 3*, *Figure 5H*). At low TFAM concentrations, we observe small punctate complexes with tracts along the DNA (*Figure 4—figure supplement 2*), which convert into large punctate complexes at higher TFAM concentrations (*Figure 4A*).

In the nuclear genome, compaction of DNA serves to protect DNA from damage as well as regulate expression, replication, and repair of DNA. Mitochondrial nucleoid compaction is believed to be associated with regulatory properties, such as expression and replication (*Brüser et al., 2021*). TFAM-induced DNA compaction has been shown to protect mtDNA from enzymatic processes used in techniques, such as Fiber-seq (*Isaac et al., 2024*); however, it is unknown if TFAM protects DNA from damage. To determine if nucleoid compaction protects DNA from UVC-induced damage, we utilized both in vitro reconstituted nucleoids and an inducible TFAM overexpression cell model to investigate whether the compactional state of mitochondrial nucleoids serves to protect the DNA from damage. In vitro, even at the highest concentration of TFAM, where the entire mtDNA appears compact in AFM images, we observe no protection of DNA from UVC (*Figure 5H and I*). Similarly, in cells where TFAM is overexpressed ~ twofold, which others have shown is associated with an increase in the number of inaccessible (presumably compacted) nucleoids (*Isaac et al., 2024*), we observe no protection from UVC-induced DNA damage or changes in accumulated DNA damage levels (*Figure 5E, F and G*). Taken together, these results indicate that TFAM does not appear to protect DNA from UVC-induced DNA damage formation. However, it remains possible that increased compaction of mtDNA by TFAM could protect DNA from other types of DNA damage that form as a result of chemical or enzymatic processes, such as reactive oxygen species or alkylating agents that require the DNA to be accessible. This may be particularly true for larger DNA-reactive chemicals, such as PAHs or mycotoxins. Finally, while TFAM overexpression in cells does not alter the levels of mtDNA damage after UVC over time, TFAM knockdown is associated with an increase in baseline mtDNA damage in cells over time (in the absence of UVC) and also results in a persistence in mtDNA damage over time after UVC exposure (*Figure 5J*). Additional work will be required to discriminate whether the persistence in measured mtDNA damage after UVC exposure in the TFAM knockdown is because of slower mtDNA removal, slower dilution via reduced mtDNA replication, increased damage by endogenous processes, or some other factor.

Despite not protecting mtDNA from UVC-induced DNA damage, our AFM and array-based binding data suggest that TFAM may act as a damage sensing protein via increased binding and its compaction-promoting function. Our AFM data show significantly increased TFAM-mediated compaction of irradiated vs unirradiated pUC19 DNA (*Figure 4*). Specifically, in the presence of irradiation, TFAM promotes compact nucleoid structures, whereas in the absence of irradiation, the TFAM clusters are more loosely associated (*Figure 4*, *Figure 4—figure supplement 3*). These results suggest that UVC irradiation promotes the cooperative assembly of TFAM on DNA. These TFAM assemblies do not appear to supercoil the DNA, but rather appear to bring distal DNA segments together, consistent with previous studies (*Figure 4*, *Figure 4—figure supplement 2*; *Kukat et al., 2015*). As such, UVC lesions could potentially serve as nucleation sites for TFAM assemblies, and these local clusters may come together to form a compact nucleoid structure containing large regions of the DNA. This suggestion is consistent with laser tweezer studies on stretched DNA that showed formation of multiple small linear clusters of TFAM that can come together to form a larger linear cluster (*Farge et al., 2012*). In our studies, the DNA is not stretched and, therefore, the TFAM assemblies can come together through space to form compact nucleoids formed via TFAM-DNA interactions that promote further oligomerization of TFAM with itself. Notably, UVC appears to significantly stabilize TFAM-TFAM interactions to promote a compact nucleoid. This increased compaction of DNA by TFAM following UVC irradiation represents a novel role for TFAM which may enable TFAM to change the architecture of mitochondrial genomes harboring DNA damage. The degree of compaction of mtDNA in vivo following DNA damage remains to be tested and will need to be assessed on a single-nucleoid level given the already heterogeneous nature of mtDNA packaging.

Our high-throughput TFAM binding data reveal a redistribution of z-scores upon UVC irradiation, with more uniform distributions of TFAM occupancy across the genomic sequences for the irradiated than unirradiated DNAs (*Figure 3*), indicating significantly reduced DNA-binding specificity. The observed reduction in sequence specificity in the context of UVC irradiation suggests that TFAM occupancy on UVC-irradiated DNA outcompetes any sequence specificity it has, at least in the absence of other proteins. These data reflect an extensively damaged genome because the DNA was irradiated at a high dose of UVC, 1500 J/m$^2$. In contrast, the DNA in the AFM experiments contains on average approximately three lesions per plasmid. Strikingly, the significant increase in compaction as evidenced by the increase in punctate structures of pUC19 by TFAM in the presence of UVC irradiation (*Figure 4D*) suggests that only a few lesions are sufficient to disrupt the DNA-binding specificity of TFAM and drive dramatic compaction of pUC19.

In our AFM images, we observe tracts of protein binding along the DNA that bridge two regions of DNA together (*Figure 4A and C*, *Figure 4—figure supplements 2 and 3*) in a zipper-like manner, without inducing any significant supercoiling. These results are consistent with previous studies (*Kukat et al., 2015*; *Kaufman et al., 2007*). In contrast, the X-ray crystal structures of TFAM bound to DNA show the protein wrapped around the DNA, forming a U-turn bend in the DNA (*Ngo et al., 2014*; *Ngo et al., 2011*; *Rubio-Cosials et al., 2011*). This mode of binding to plasmid DNA would induce supercoiling (*van Noort et al., 2004*). Taken together, these results indicate that TFAM can bind DNA in at least two different states, one which induces a U-turn bend in the DNA and one that does not dramatically distort the DNA. In the AFM images, in general, we observe a single cluster of TFAM per plasmid, with or without adjacent fiber formation. In the absence of UV irradiation, the clusters occupy a smaller region of DNA and exist in looser, more intermediate states (*Figure 4D*), whereas in the irradiated DNA, TFAM forms more punctate clusters that contain more of the plasmid DNA (*Figure 4D*). One possible explanation for how only a few lesions could promote increased compaction of the DNA by TFAM is that TFAM may preferentially bind UV damage, which distorts DNA, in the U-turn mode. This U-turn in the DNA would bring two DNA double strands close to one another and, therefore, promote local zipping up of the DNA. Subsequently, these clusters may cooperatively assemble together to form tight compact structures (*Figure 4C and E*).

## Conclusions and future directions

Overall, we found that cellular responses to mtDNA damage, including increased mtDNA degradation, upregulation of mtDNA replication machinery, upregulated TFAM, and increased mtRNA transcription occur in the absence of loss of mitochondrial membrane potential or changes to ATP levels. We identified TFAM as a potential DNA damage sensing protein in that it promotes UVC-dependent

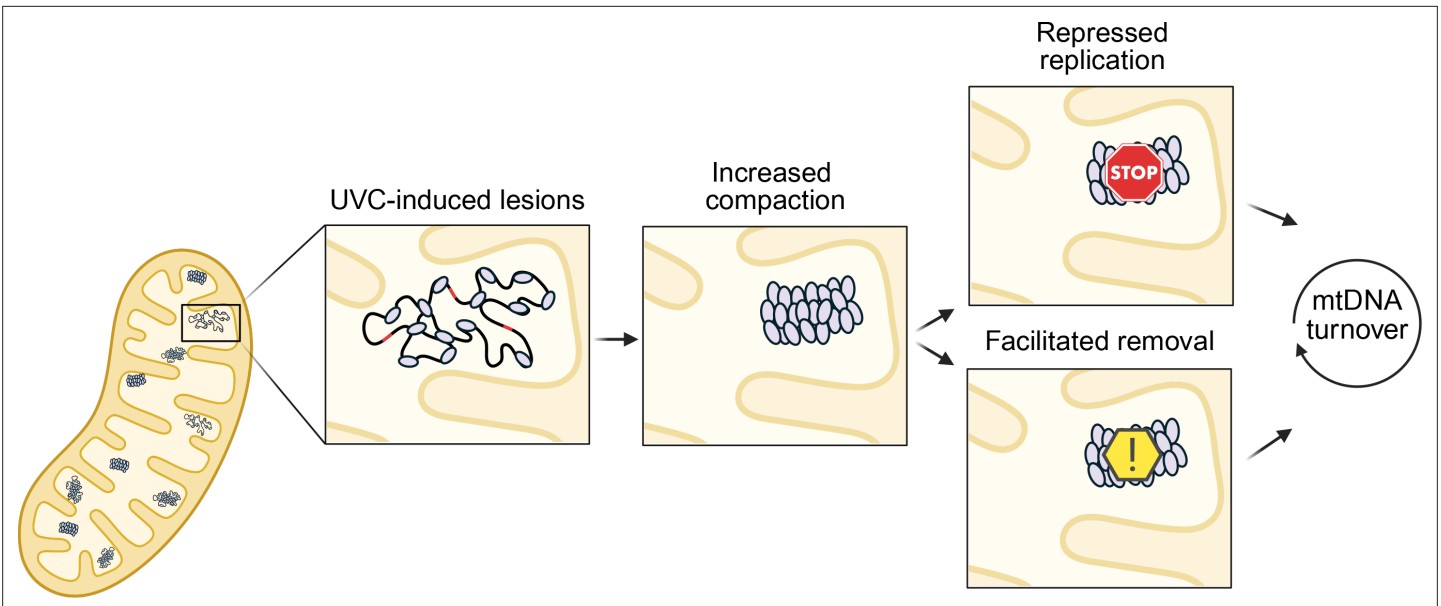

**Figure 6.** Hypothesized cellular model of the role of Transcription Factor A, Mitochondrial (TFAM) in mitochondrial DNA (mtDNA) damage sensing. The in vitro data provided in this study indicate that TFAM more readily compacts DNA harboring UV-induced lesions. While there is currently no in vivo evidence to suggest damaged mtDNA is more compacted, future work should determine whether this phenomenon is occurring in cells. In vivo, this feature may serve to 'tag' mitochondrial genomes as damaged, which could lend itself to repression of the replication of damaged genomes, flagging damaged genomes for targeted degradation, or both. Both active removal and repression of replication would allow for removal of damaged genomes during mtDNA turnover processes and provide a mechanism for preventing mtDNA mutagenesis. Created with BioRender.com.

conformational changes in the nucleoids in vitro, making them more compact. The increased compaction may serve to signal damage in the mitochondrial genome. For example, once 'tagged' by TFAM, damaged mtDNAs might be actively removed via mitophagy (*Bess et al., 2012*; *Leuthner et al., 2022*), autophagy (*Bess et al., 2013*), extracellular export, TFAM-mediated nucleoid-phagy (*Liu et al., 2024*), or by more recently identified non-autophagic pathways (*Sen et al., 2022*; *Newman et al., 2024*). Indeed, we found that TFAM knockdown alters the levels of mtDNA damage over time. Alternately, simply by removing these damaged genomes from the pool that is actively used for transcription and replication by making them inaccessible to other proteins, TFAM-mediated compaction could serve to store damaged genomes until they are removed passively by normal mtDNA turnover processes. In any case, these non-repair pathways for removing mtDNA damage may provide a mechanism for preventing mtDNA mutagenesis. These possibilities are illustrated schematically in *Figure 6*, which depicts a hypothetical model that will require additional work to test in vivo. Overall, this work presents a first step towards understanding how TFAM may play a role in sensing and responding to irreparable mtDNA damage.

## Materials and methods
### Cell culture
HeLa cells were kindly provided by Dr. Chantell Evans (Duke University). TFAM-HA TetOn HeLa S3 cells were kindly provided by Dr. Stefan Isaac and Dr. Stirling Churchman (Harvard University). Cells were grown in DMEM containing glucose and pyruvate (Thermo Fisher Scientific 11995073) supplemented with 10% Fetal Bovine Serum (Thermo Fisher Scientific 10437028), 1% penicillin/streptomycin and maintained at 37 °C with 5% $CO_2$ in a humidified chamber. Cells were routinely tested for mycoplasma contamination and authenticated using STR profiling by the Duke University Cell Culture Facility.

### UVC exposures in cell culture
HeLa cells were grown to 50% confluency. At the time of exposure, media was removed, cells were washed once in PBS, aspirated to remove as much of the PBS as possible to facilitate the effects of the

UVC irradiation, and then exposed to UVC using an ultraviolet lamp with built-in UVC sensor (CL-1000 Ultraviolet Crosslinker, UVP, Upland, CA, USA) with peak emission at 254 nm. Following exposure (~1 min), cell culture media was immediately replaced.

## Cell viability

To assess cell viability, 10,000 cells were plated in black clear-bottom 96-well plates and allowed to adhere overnight. Cells were washed with PBS and then dosed with UVC as described in the previous section. Experiments were performed in triplicate with three biological replicates completed for all doses. The 0-, 24-, and 48 hr dose responses were analyzed at their respective time points. Cell viability was assessed using the alamarBlue HS Cell Viability Reagent (Invitrogen) and a FLUOstar Optima microplate reader. The data was analyzed via two-way ANOVA with a Dunnett's correction for multiple comparisons to the control for each time point.

## Gene expression

Quantitative real-time RT-PCR was performed for *TFAM, POLG, NRF-1, PGC1a, POLRMT*, and *ND-1*. RNA was extracted from HeLa cells using an RNeasy Kit (QIAGEN) and stored at –80 °C until analysis. qPCR was performed using the qTower$^3$ Real-Time PCR System (Analytik-Jena). The *GAPDH* Endogenous Control (Taqman probe ID: Hs02786624_g1, VIC-MGB) and either *TFAM* (Taqman probe ID: Hs00273372_s1, FAM-MGB), *POLG* (Taqman probe ID: Hs00160298_m1, FAM-MGB), *NRF-1* (Taqman probe ID: Hs00602161_m1, FAM-MGB), *PGC1a* (Taqman probe ID: Hs00173304_m1, FAM-MGB), *POLRMT* (Taqman probe ID: Hs04187596_g1, FAM-MGB), or *ND-1* (Taqman probe ID: Hs02596873_s1, FAM-MGB) probes were multiplexed in a 1:1 ratio for each sample. Results for *GAPDH* were used to normalize for RNA input. PCR reactions each contained 100 ng RNA, 0.5 µL of each probe, 5 µl of Quantabio qScript one-step RT-qPCR ToughMix (QuantaBio) and was brought to a total reaction volume of 10 µL with nuclease-free water. Samples were run in triplicate, and PCR cycling conditions were as follows: 50 °C for 10 min, followed by 95 °C for 1 min, followed by 40 cycles of 95 °C for 10 s and 60 °C for 1 min. To examine the expression of each gene, the delta delta CT method was used to calculate fold change in gene expression. The data were analyzed via two-way ANOVA with Dunnett's post-hoc test for multiple comparisons.

## Mitochondrial membrane potential

HeLa cells were plated and allowed to attach and proliferate for 24 hr. Then, cells were exposed to either 0, 10, 30, or 50 J/m$^2$ of UVC. Cells were harvested and stained to acquire by flow cytometry at 6 and 24 hr post UVC exposure. Each UVC dose had a tetramethylrhodamine, methyl ester (TMRM)-stained sample and an unstained control. Non-UVC-exposed cells from each time point had additional TMRM-stained tubes for carbonyl cyanide-p-trifluoromethoxyphenylhydrazone (FCCP) positive controls. Cells were aliquoted ($4.0 \times 10^5$ per sample) and washed with PBS. Samples were stained in 1 mL of Live Cell Imaging Medium with a final concentration of 15 nM TMRM for 20 min. TMRM was left in cell suspension during flow acquisition. Samples were acquired on a BD FACSCanto-II flow cytometer using the blue laser (488 nm) and the 585/42 detection filter to excite and detect TMRM. Events were gated to include live, single cells, and 10,000 events were acquired from this population. Immediately before their acquisition, TMRM-stained control cells were spiked with 5 µM FCCP. The FCCP positive control was incubated for 5 min before acquiring. In FlowJo, live cells were manually gated in a FSC-A vs SSC-A plot, then single cells were gated in a FSC-A vs FSC-H plot. Median fluorescence intensity (MFI) was tabulated for the histogram of each sample. Background fluorescence from the unstained sample of each UVC dose was subtracted from the corresponding stained samples. The average and standard error of three experiments were plotted. Data was analyzed via two-way ANOVA for treatment and time point, with a Dunnett's correction for multiple comparisons to the control for each time point.

## ATP quantification

To determine ATP concentrations, 10,000 HeLa cells were plated in white, flat-bottom 96-well plates. Cells were exposed to UVC irradiation as described above. At 6 or 24 hr post-UVC exposures, the Cell-Titer-Glo assay (Promega) was performed according to the manufacturer's instructions. Briefly, plates were equilibrated to room temperature before addition of 100 µL of reconstituted assay reagent. Cells

were lysed via orbital shaking (200 rpm for 2 min) followed by a 10 min incubation at room temperature. Luminescence was measured on a FLUOstar Optima microplate reader. A standard curve was prepared fresh before each assay with ATP disodium salt (Sigma) in complete DMEM. The data was analyzed via two-way ANOVA with a Dunnett's correction for multiple comparisons to the control for each time point.

### Live cell imaging

HeLa cells ($1.5 \times 10^3$) were seeded onto 35 mm glass bottom well petri dishes (Mattek) and allowed to adhere. After 48 hr in culture, cells were washed once with PBS and then exposed to either 0 or 10 J/$m^2$ UVC as described previously. Cells were then immediately incubated with Picogreen at a concentration of 1 µL/mL of medium for 1.5 hr to stain mtDNA. Cells were then washed three times and given fresh media. Twenty-four hours after the UVC exposure, cells were incubated with lysotracker red (75 nM), mitotracker deep red (150 nM), and 7 µg/mL Hoescht 33342 for 30 min, again followed by three quick washes, and then placed in imaging medium. Cells were imaged in an environmental chamber at 37 °C in an Andor Dragonfly 505 unit with Borealis illumination spinning disk confocal 100 x/1.40–0.70 HCX PL APO (Leica 11506210) oil-immersion objective and an Andor iXon Life 888 1024×1024 EMCCD camera.

### Confocal image analysis

Images were deconvoluted using Huygens SVI deconvolution software. For each channel, images were segmented in Labkit (*Arzt et al., 2022*). Segmented images were used to quantify the area of the cell, area of mitochondria, mitochondrial morphology parameters, number of Picogreen spots, and number of lysosome spots. To assess lysosome colocalization, object-oriented colocalization (*Lachmanovich et al., 2003*) was performed on segmented images using a custom Python script. At least 30 cells in each treatment group were analyzed from three independent biological replicates. The data were analyzed via two-tailed unpaired t-test.

### Quantification of DNA damage

To determine the lesion frequency associated with mitochondrial DNA, a quantitative long-range PCR assay was used (*Furda et al., 2014*). For cell samples, DNA was isolated from frozen cell pellets using genomic tips (Qiagen). QPCR was run on the samples to generate a long product (~10 kb) as well as a short product (~200 bp). Given the ability for DNA damage to block the DNA polymerase used in this PCR, the amount of damage is then calculated based on the amount of amplification. The long product is used to calculate reduced amplification compared to the controls, whereas the short product is used to normalize sample to sample variability in mitochondrial DNA copy number. The data were analyzed via two-way ANOVA with a Dunnett's post-hoc test for multiple comparisons.

### TFAM overexpression

TFAM-HA TetOn HeLa S3 cells were grown in six-well dishes to a confluency of 50% and treated with 100 ng/mL doxycycline for 48 hr. Cells were provided fresh media every 24 hr for the duration of the experiment. UVC exposure was performed as described above with cells exposed to either 0, 10, 30, 50, 70, or 100 J/$m^2$. mtDNA damage was quantified at 0 hr, 24 hr, and 48 hr following UVC exposure as described previously and cell viability was assessed. TFAM overexpression levels in control samples were confirmed by real-time PCR and western blot analysis. Real-time PCR to assess *TFAM* gene expression was performed as described above.

For western blot analysis, cells were washed twice in ice-cold PBS and then lysed using RIPA buffer containing a protease inhibitor cocktail. Protein content was determined using a BCA Assay (Pierce). Protein lysate (30 µg/well) was loaded into a 10% SDS-PAGE gel. Gels were transferred onto activated PVDF membrane, followed by immunoblot. Densitometry was performed using ImageJ. TFAM was normalized to β-actin and samples were compared between doxycycline treatment groups at their respective time points. Antibodies used were anti-TFAM antibody (Santa Cruz sc-376672) and an anti-β-Actin antibody (Sigma A5441). The data were analyzed via two-way ANOVA.

## TFAM knockdown

HeLa cells were reverse transfected in 6-well plates with 8.3 pmol siRNA (TFAM Flexitube siRNA, GeneGlobe ID SI04988487; or AllStars Negative Control siRNA, Qiagen) and 2.5 µL Lipofectamine RNAiMAX (Invitrogen) per well according to the manufacturer's instructions. Cell plating densities were optimized to avoid confluency for each time point and scaled up to harvest sufficient cells for analysis. At 24 hr post-transfection, cells were exposed to either 0 J/m$^2$ or 30 J/m$^2$ UVC and harvested immediately (time point 0 hr) or at 24 or 48 hr post-UVC. Harvested cells were washed with PBS, pelleted, and frozen at –80 °C prior to DNA extraction. Data was analyzed via three-way ANOVA with a Dunnett's post-hoc test for multiple comparisons.

## Purification of human TFAM

Human_TFAM_NoMTS_pET28 was a gift from David Chan (Addgene plasmid # 34705; http://n2t. net/addgene:34705; RRID:Addgene_34705). Recombinant human TFAM was purified as previously described (*Ngo et al., 2011*), flash frozen in liquid nitrogen, and stored at –80 °C. Protein concentration was determined using a Qubit Protein BR Assay (Thermo A50668).

## Generation of substrates for atomic force microscopy

To assess alterations in TFAM compaction of DNA in the context of UVC-induced lesions, we assessed TFAM binding to circular pUC19. Levels of UVC exposure were chosen based on previous dose response curves performed on naked mtDNA indicating a linear increase in lesions associated with increasing UVC exposure, where 10 J/m$^2$ results in roughly 1 lesion per 10 kb (*Colton et al., 2014*). Given the similar content of adjacent pyrimidine bases (whole mitochondrial genome: 535 /kb; 8.9 kb product used in mtDNA damage assay: 532 /kb; and pUC19: 510 /kb), we exposed the pUC19 (2.686 kb in size) to 100 J/m$^2$ to induce roughly three lesions per DNA molecule.

## Atomic force microscopy

Topoisomerase I (NEB # M0301S) was used to relax all pUC19 DNA substrates used for AFM TFAM protein-DNA experiments. Each reaction was carried out by using 1 µg of DNA (either UV-irradiated or undamaged), 10 x CutSmart Buffer, and 1 unit of Topoisomerase I diluted to a total reaction volume of 50 µL. The reaction was then placed in a thermocycler and run at 37 °C for 15 min, and afterwards a PCR cleanup kit (QIAGEN) was used to remove protein and other impurities before performing AFM experiments.

Undamaged or damaged DNA (1.5 ng/µL) was incubated for 2 min on ice with varying concentrations of TFAM in 25 mM HEPES, 10 mM Mg(OAc)$_2$, 25 mM NaOAc, 75 mM K(OAc), pH 7.5, then deposited onto ethanolamine-treated mica and imaged in air using a JPK NanoWizard 4 XP in tapping mode. Mica was treated with ethanolamine by vapor deposition in a desiccator by aliquoting 20 µL of ethanolamine stock onto a piece of parafilm and placing it into the desiccator with freshly peeled mica for 15 min to allow time for the ethanolamine to vaporize and transfer onto the mica surface. Cantilevers used were PPP-NCHR (Nanoworld, Matterhorn, Switzerland) with nominal resonance frequencies of 330 kHz. Images were taken at a 2.0×2.0 µm area for small DNA fragments, such as pUC19, while the full-length mitochondrial DNA images were between 2–5 µm. Images of TFAM and small DNA oligonucleotides were taken at a 1.0×1.0 µm area on untreated mica. All AFM images were taken with a resolution of 512×512 pixels and a scan rate of 2 Hz. Additionally, each TFAM-DNA complex with pUC19 was categorized into three different categories: dispersed, intermediate, or punctate, based on the degree of observable DNA compaction and height of the TFAM nucleoid structure.

Images were analyzed using the MATLAB program ImageMetrics (https://imagemetrics.wordpress. com/) in MATLAB version R2020b (MathWorks, Natick MA). Images were line-wise flattened, and subsequently, a threshold was applied to isolate the individual protein only or protein-DNA complexes on the surface. For each of these complexes or proteins, the volume, area, and height were recorded. Histograms of the volumes of TFAM protein alone or in the presence of DNA were generated from the data using MATLAB version R2020b.

We used AFM to determine the oligomerization state of TFAM bound to some DNA oligonucleotides (33 base pairs) used for the fluorescence anisotropy studies. TFAM (30 nM) was incubated on ice in the absence or presence of DNA (10 nM) for 2 min, deposited on the surface, rinsed, and dried with N$_2$. ImageMetrics was used to determine the volumes of the individual particles on the surface

as described earlier. The AFM volume of proteins are directly proportional to the protein molecular weight allowing determination of the oligomerization state (*Ratcliff and Erie, 2001*; *Yang et al., 2003*; *Chelico et al., 2008*; *Brar et al., 2008*). The oligomerization states are determined from histogram plots of the distribution of volumes that were generated in MATLAB version R2020b.

## In vitro nucleoid reconstitution

Full-length linear mtDNA was generated by long-range PCR amplification using the following primers; F, 5′ – GGT TCA GCT GTC TCT TAC TTT TAA CCA GTG – 3′ and R, 5′ – CTC GTG GAG CCA TTC ATA CAG GTC – 3′. Genomic DNA was isolated from HeLa cells using the Qiagen QiAmp DNA Mini Kit according to the manufacturer's instructions and used as template DNA. PCR reactions were carried out using 100 ng template in 50 µL reactions with LongAmp Hot Start Taq Polymerase (New England Biosciences). DNA was amplified using a two-step amplification protocol: 95 °C for 2 min, 30 cycles of 95 °C for 20 s and 68 °C for 13 min, and a final extension at 68 °C for 18 min. PCR product was then purified using a Zymo Genomic DNA Clean & Concentrator kit (Zymo Research D4011) and the DNA concentration was determined using a Nanodrop One (Thermo Scientific).

In vitro nucleoids were generated in 50 µL reactions, containing 20 ng/µL full-length linear mtDNA with 0, 100, 250, or 1000 nM TFAM (added last) in binding buffer (25 mM HEPES pH 7.5, 75 mM KOAc, 25 mM NaOAc, 10 mM Mg(OAc)$_2$). The degree of compaction of the mtDNA at these TFAM concentrations was confirmed using AFM as described earlier. After the addition of TFAM, reactions were incubated on ice for 2 min, then deposited as a 50 µL drop onto a petri dish lid and then immediately exposed to either 0, 5, 10, 30, or 50 J/m$^2$ UVC. Following UVC treatment, a Zymo Genomic DNA Clean & Concentrator 10 kit (Zymo Research D4011) was used to remove the protein, and the DNA concentration was determined using a Nanodrop One (Thermo Scientific). Samples were diluted to 25 pg/µL and levels of mtDNA damage were quantified as described above.

## High-throughput, high-resolution binding of TFAM to the mitochondrial genome

High-density DNA chips (or arrays) printed with oligos covering the entire mitochondrial genome at a 2 bp resolution were used to measure TFAM binding in the presence and absence of UV-induced damage. Single-stranded DNA arrays were purchased from Agilent in a 4×180 k format containing four identical chambers with 176,024 DNA spots in each chamber. The DNA library synthesized on the array contained the entire mitochondrial genome in 33-nt variable sequences and a 27-nt primer binding sequence. A sliding window approach was used to partition the hg38 human mitochondrial genome into overlapping 33-nt regions, where each sequential window was shifted by 2 nt along the mtDNA sequence (the single 'N' was replaced with a cytosine). For every mtDNA window, an additional DNA sequence was generated using the reverse complement of the window, such that the full genome was synthesized de novo in both the 5′ to 3′ and 3′ to 5′ orientations on the glass slide. TFAM binding to a universal DNA library containing all possible 7-mers (*Berger et al., 2006*) was performed prior to library designs to determine the range of binding specificity, optimize the final protein concentration to be used, and select low-affinity non-mitochondrial probes to include in the sequence library for normalization purposes. The number of replicate spots for each sequence on the array was 10. In total, there were 8285 unique probes in each orientation on the array (16,570 total), corresponding to the mitochondrial genome.

The DNA was double-stranded on the arrays via primer extension, as described previously (*Berger et al., 2006*), and then subjected to 1500 J/m$^2$ UVC irradiation to damage DNA in two of the chambers on the array, as previously described (*Mielko et al., 2023*). Finally, purified TFAM as well as Penta-His Alexa 488 (Qiagen) were incubated in the chambers in TFAM binding buffer (25 mM HEPES, 10 mM Mg(OAc)$_2$, 100 mM NaOAc, pH 7.5) as previously described (*Berger and Bulyk, 2009*) and the fluorescent signal associated with bound protein for each DNA spot was determined using a GenePix 4400 A microarray scanner and the GenePix Pro 7.3 software. The assay was performed using a final concentration of either 30 nM or 300 nM TFAM. In total, assays were run at 30 nM TFAM with and without UVC, as well as 300 nM TFAM with and without UVC.

Fluorescence intensity values for each spot were normalized via detrending, as previously described (*Berger and Bulyk, 2009*). Mean, median, standard deviation, and standard error were calculated from normalized fluorescence intensity values across the 10 replicates for each spot. Kernel density

estimates were calculated for the fluorescence values of the bottom universal binders for the conditions in each chamber, and a Gaussian curve was fit to those data. For each chamber, the mean and standard deviation estimates from the Gaussian fit were used to convert all median fluorescence intensity values into TFAM binding z-scores, which reflect the DNA binding specificity of TFAM under that experimental condition. Given that the assay was performed on sequences generated in 5' to 3' orientation as well as a 3' to 5' orientation, the maximum z-score obtained from each orientation (*i.e.* whichever was higher) was used to generate binding plots. Plots were then smoothed using a 22-nt window, in order to account for the 22-nt footprint of TFAM binding. Most binding z-scores are expected to be low, regardless of the binding conditions, as most sequences are expected to show some low background level of binding. However, under conditions where TFAM binds DNA with high specificity, we expect the array-based assay to also identify sequences with high z-score, *i.e.*, sequences that TFAM can bind with higher occupancy, and distinguish them from background. In contrast, conditions where all z-scores are generally low are indicative of low binding specificity throughout, as all sequences are bound similarly by TFAM, and no particular target sites stand out. All statistics were performed using a custom Python notebook.

## Fluorescence polarization

Sequences representing a broad range of z-scores from the array-based experiments (**Supplementary file 2**) were ordered from Integrated DNA Technologies as HPLC-purified, double-stranded oligos (reverse complement not shown in table) with a 5' 6-carboxyfluorescein modification on the strand listed. Each sequence was tested with and without 1500 J/m$^2$ UVC irradiation to match conditions used in the array-based assay. Each condition was tested in triplicate.

TFAM was serially diluted in binding buffer (25 mM HEPES, 10 mM Mg(OAc)$_2$, 25 mM NaOAc, 75 mM K(OAc), pH 7.5) from 15 to 0 nM on ice. Dilutions were then mixed in a low-binding Costar 96-well half-area black well plate (Corning, Corning, NY) with DNA diluted in binding buffer, for a final DNA concentration of 1 nM and volume of 55 µL in each well. Plates were read with a PHERAstar microplate reader from BMG Labtech (Cary, NC) using an FP filter with excitation at 485 nm and emission at 590 nm. Anisotropy was calculated with **Equation 1**:

$$r = \frac{I_{\parallel} - I_{\perp}}{I_{\parallel} + 2I_{\perp}} \qquad (1)$$

where r is anisotropy and $I_{\parallel}$ and $I_{\perp}$ are fluorescence intensity in the parallel and perpendicular direction, respectively. Anisotropy was then plotted against TFAM concentration and fit to **Equation 2**:

$$r = r_f \frac{r_b \cdot [\text{TFAM}]^n}{K_D^n + [\text{TFAM}]^n} \qquad (2)$$

where $r_f$ is the fraction of signal from free DNA, $r_b$ is the fraction of signal from bound DNA, [TFAM] is TFAM concentration, n is the Hill coefficient, and $K_D$ is the dissociation constant. The variance $\sigma^2$ associated with each of the derived $K_D$ and n values from individual fits was identified from the diagonal of the covariance matrix, and the inverse-variance weighted mean $\bar{x}_{wtd}$ and standard error $\left(\sigma_{\bar{x}}\right)_{wtd}$ were calculated using **Equations 3; 4**, respectively, across the three experimental replicates:

$$\bar{x}_{wtd} = \frac{\sum\limits_{i=1}^{m} x_i/\sigma_i^2}{\sum\limits_{i=1}^{m} 1/\sigma_i^2} \qquad (3)$$

$$\left(\sigma_{\bar{x}}\right)_{wtd} = \sqrt{\frac{\left(\frac{\sum\limits_{i=1}^{m} x_i^2/\sigma_i^2}{\sum\limits_{i=1}^{m} 1/\sigma_i^2} - \bar{x}_{wtd}^2\right)\left(\frac{m}{m-1}\right)}{m}} \qquad (4)$$

where $x_i$ is the derived value from each fit, $m$ is the number of replicates in the condition (*James, 2020*). Curve fitting and statistical analysis were performed in Python.

## Acknowledgements

We thank the UNC Macromolecular Interactions Facility for providing instrumentation for fluorescence anisotropy as well as the Duke Light Microscopy Core for providing access to microscopes and image analysis software. This work was supported by the National Science Foundation award GRFP DGE-1644868 to DEK, NIEHS P42ES010356 to JNM, NIEHS T32ES021432 to JNM, NIGMS R35 GM127151 to DE, NSF/MCB 2324614 to RG, and P30CA016086 to the UNC Center for Structural Biology. The content is solely the responsibility of the authors and does not necessarily represent the official views of the National Institutes of Health.

## Additional information

### Funding

| Funder | Grant reference number | Author |
| --- | --- | --- |
| National Institute of Environmental Health Sciences | P42ES010356 | Joel Meyer |
| National Institute of Environmental Health Sciences | T32ES021432 | Joel Meyer |
| U.S. National Science Foundation | GRFP DGE-1644868 | Dillon E King |
| National Institute of General Medical Sciences | R35 GM127151 | Dorothy A Erie |
| U.S. National Science Foundation | 2324614 | Raluca Gordân |
| National Institute of Environmental Health Sciences | R35ES035049 | Joel Meyer |

The funders had no role in study design, data collection and interpretation, or the decision to submit the work for publication.

### Author contributions

Dillon E King, Conceptualization, Formal analysis, Investigation, Writing – original draft; Emily E Beard, Formal analysis, Investigation, Writing – original draft; Matthew J Satusky, Formal analysis, Investigation, Visualization, Writing – original draft; Alex George, Ian Ryde, Caitlin Johnson, Emma L Dolan, Yuning Zhang, Wei Zhu, Hunter Wilkins, Evan Corden, Formal analysis, Investigation; Susan K Murphy, Supervision, Funding acquisition; Dorothy A Erie, Raluca Gordân, Joel Meyer, Conceptualization, Supervision, Funding acquisition, Writing – original draft

### Author ORCIDs

Dillon E King  https://orcid.org/0000-0002-1685-5252
Emily E Beard  https://orcid.org/0000-0001-8660-6094
Matthew J Satusky  https://orcid.org/0000-0003-2089-6935
Alex George  https://orcid.org/0000-0002-2802-2565
Ian Ryde  https://orcid.org/0009-0005-6773-7335
Emma L Dolan  https://orcid.org/0000-0003-1086-3275
Yuning Zhang  https://orcid.org/0000-0002-2967-4207
Hunter Wilkins  https://orcid.org/0000-0003-4903-0043
Evan Corden  https://orcid.org/0000-0002-6900-4662
Susan K Murphy  https://orcid.org/0000-0001-8298-7272
Dorothy A Erie  https://orcid.org/0000-0003-0363-4693

Raluca Gordân [ORCID] https://orcid.org/0000-0002-6404-6556
Joel Meyer [ORCID] https://orcid.org/0000-0003-1219-0983

Reviewer #1 (Public review): https://doi.org/10.7554/eLife.108862.3.sa1
Reviewer #2 (Public review): https://doi.org/10.7554/eLife.108862.3.sa2
Reviewer #3 (Public review): https://doi.org/10.7554/eLife.108862.3.sa3
Author response https://doi.org/10.7554/eLife.108862.3.sa4

## Additional files

### Supplementary files

Supplementary file 1. Total number of Transcription Factor A, Mitochondrial (TFAM) complexes analyzed in *Figure 4D*.

Supplementary file 2. Sequences used for Transcription Factor A, Mitochondrial (TFAM) $K_D$ measurements.

MDAR checklist

### Data availability

All the raw and processed protein-DNA binding assay data generated and used in this study are located on NCBI Gene Expression Omnibus (GEO) [GSE281949]. Custom Python notebooks are available on GitHub (copy archived at *Satusky, 2026*). All other data are available in the Duke Research Data Repository (https://doi.org/10.7924/r40s0211r).

The following datasets were generated:

| Author(s) | Year | Dataset title | Dataset URL | Database and Identifier |
|---|---|---|---|---|
| King DE, Beard EE, Satusky MJ, Ryde I, George A, Johnson C, Dolan EL, Zhang Y, Zhu W, Wilkins H, Corden E, Murphy SK, Erie D, Gordan R, Meyer JN | 2026 | UV irradiation alters TFAM binding specificity and compaction of DNA | https://www.ncbi.nlm.nih.gov/geo/query/acc.cgi?acc=GSE281949 | NCBI Gene Expression Omnibus, GSE281949 |
| George A, Zhu W, Gordân R, Erie D, Murphy S, Satusky M, Beard E, Corden E, Meyer J, King D, Dolan E, Ryde I, Wilkins H, Johnson C, Zhang Y | 2026 | Data from: UV irradiation alters TFAM binding specificity and compaction of mitochondrial DNA | https://doi.org/10.7924/r40s0211r | Duke Research Data Repository, 10.7924/r40s0211r |

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
