## [Editor Report · eLife Assessment]

Mitochondrial DNA (mtDNA) exhibits a degree of resistance to mutagenesis under genotoxic stress, and this study on the mitochondrial Transcription Factor A (TFAM) presents **important** data concerning the possible mechanisms involved. The presented data are **solid**, technically rigorous, and consistent with established literature findings. The experiments are well-executed, providing **convincing** evidence on the change of TFAM-DNA interactions following UVC irradiation.

---

## [Referee Report · Reviewer #1 (Public review)]

Summary:

The authors investigate how UVC induced DNA damage alters the interaction between the mitochondrial transcription factor TFAM and mtDNA. Using live-cell imaging, qPCR, atomic force microscopy (AFM), fluorescence anisotropy, and high-throughput DNA-chip assays, they show that UVC irradiation reduces TFAM sequence specificity and increases mtDNA compaction without protecting mtDNA from lesion formation. From these findings the authors suggest that TFAM acts as a "sensor" of damage rather than a protective or repair-promoting factor.

Strengths:

(1) The focus on UVC damage offers a clean system to study mtDNA damage sensing independently of more commonly studied repair pathways, such as oxidative DNA damage. The impact of UVC damage is not well understood in the mitochondria and this study fills that gap in knowledge.

(2) In particular, the custom mitochondrial genome DNA chip provides high resolution mapping of TFAM binding and reveals a global loss of sequence specificity following UVC exposure.

(3) The combination of in vitro TFAM DNA biophysical approaches combined with cellular responses (gene expression, mtDNA turnover) provides a coherent multi-scale view.

(4) The authors demonstrate that TFAM induced compaction does not protect mtDNA from UVC lesions, an important contribution given assumptions about TFAM providing protection.

Weaknesses:

(1) The authors show a decrease in mtDNA levels and increased lysosomal colocalization but do not define the pathway responsible for degradation. Distinguishing between replication dilution, mitophagy, or targeted degradation would strengthen the interpretation and justifies future experiments.

(2) The manuscript briefly notes enrichment of TFAM at certain regions of the mitochondrial genome but provides little interpretation of why these regions are favored. Discussion of whether high-occupancy sites correspond to regulatory or structural elements would add valuable context.

(3) The authors provide a discrepancy between the anisotropy and binding array results. The reason for this is not clear and one wonders if an orthogonal approach for the binding experiments would elucidate this difference (minor point).

Assessment of conclusions:

The manuscript successfully meets its primary goal of testing whether TFAM protects mtDNA from UVC damage and the impact this has on the mtDNA. While their data points to an intriguing model that TFAM acts as a sensor of damaged mtDNA, the validation of this model requires further investigation to make the model more convincing. This is likely warranted for a followup study. Also the biological impact of this compaction, such as altering transcription levels is not clear in this study.

Impact and utility of the methods:

This work advances our understanding of how mitochondria manage UVC genome damage and proposes a structural mechanism for damage "sensing" independent of canonical repair. The methodology, including the custom TFAM DNA chip, will be broadly useful to the scientific community.

Context: The study supports a model in which mitochondrial genome integrity is maintained not only by repair factors, but also by selective sequestration or removal of damaged genomes. The demonstration that TFAM compaction correlates with damage rather than protection reframes an interesting role in mtDNA quality control.

Comments on revised version:

The authors addressed all concerns during the revision.

---

## [Referee Report · Reviewer #2 (Public review)]

Summary:

King et al. present several sets of experiments aimed to address potential impact of UV irradiation on human mitochondrial DNA as well as possible role of mitochondrial TFAM protein in handling UV irradiated mitochondrial genomes. The carefully worded conclusion derived from the results of experiments performed with human HeLa cells, in vitro small plasmid DNA, with PCR-generated human mitochondrial DNA and with UV-irradiated small oligonucleotides is presented in the title of the manuscript: "UV irradiation alters TFAM binding to mitochondrial DNA". Authors also interpret results of somewhat unconnected experimental approaches to speculate that "TFAM as a potential DNA damage sensing protein in that it promotes UVC-dependent conformational changes in the [mitochondrial] nucleoids, making them more compact. They further propose that such a proposed compaction might trigger removal of UV-damaged mitochondrial genomes as well as facilitates replication of undamaged mitochondrial genomes.

Strengths:

(1) Authors presented convincing evidence that a very high dose (1500 J/m2) of UVC applied to oligonucleotides covering the entire mitochondrial DNA genome alleviates sequence specificity of TFAM binding (Figure 3). This high dose was sufficient to cause UV-lesions in a large fraction of individual oligonucleotides. The method has been developed in the lab of one of the corresponding authors (ref. 74) and is technically well refined. This result can be published as is or in combination with other data.

(2) Manuscript also presents AFM evidence (Figure 4) that TFAM, which was long known to facilitate compaction of mitochondrial genome (Alam et al., 2003; PMID 12626705 and follow up citations), causes in vitro compaction of a small pUC19 plasmid and that approximately 3 UVC lesions per plasmid molecule results in slight albeit detectable increase in TFAM compaction of the plasmid.

Both results are discussed in line of a possible extrapolation to in vivo phenomena. The revised version of the discussion includes a clear statement that no in vivo support was provided within the set of experiments presented in the manuscript.

Weaknesses:

The experiments presented on Figures 3 and 4 may support the speculation that TFAM can carry protective role of eliminating mitochondrial genomes with bulky lesions by way of excessive compaction and removal damaged genomes from the in vivo pool, however extensive additional studies that would go well beyond the experiments described in this paper are needed to fill the gap between this set of results and the proposed explanations.

---

## [Referee Report · Reviewer #3 (Public review)]

Summary

The study is grounded in the observation that mitochondrial DNA (mtDNA) shows some resistance to mutagenesis under genotoxic stress. The manuscript focuses on the effects of UVC-induced DNA damage on TFAM-DNA binding in vitro and in cells. The authors demonstrate increased TFAM-DNA compaction following UVC irradiation in vitro, as assessed by high-throughput protein-DNA binding assays and atomic force microscopy (AFM). The authors did not observe a similar trend in fluorescence polarization assays and attributed the difference in the extent of TFAM oligomerization as a potential reason. In cells, the authors found that UVC exposure increased mRNA levels of TFAM, POLG, and POLRMT without altering mitochondrial membrane potential. Overexpressing TFAM in cells or varying TFAM concentration in reconstituted nucleoids did not alter the accumulation or disappearance of mtDNA damage. Based on their data, the authors proposed a plausible model: following UVC-induced DNA damage, TFAM facilitates nucleoid compaction, which may signal damage in the mitochondrial genome. The proposed model may inspire future follow-up studies to further study the role of TFAM in sensing UVC-induced damage.

Comments on revised version:

The authors have addressed the reviewer's concerns.

---

## [Author Response]

The following is the authors’ response to the original reviews.

**Public Reviews:**

**Reviewer #1 (Public review):**
Summary:The authors investigate how UVC-induced DNA damage alters the interaction between the mitochondrial transcription factor TFAM and mtDNA. Using live-cell imaging, qPCR, atomic force microscopy (AFM), fluorescence anisotropy, and high-throughput DNA-chip assays, they show that UVC irradiation reduces TFAM sequence specificity and increases mtDNA compaction without protecting mtDNA from lesion formation. From these findings, the authors suggest that TFAM acts as a "sensor" of damage rather than a protective or repair-promoting factor.Strengths:(1) The focus on UVC damage offers a clean system to study mtDNA damage sensing independently of more commonly studied repair pathways, such as oxidative DNA damage. The impact of UVC damage is not well understood in the mitochondria, and this study fills that gap in knowledge.(2) In particular, the custom mitochondrial genome DNA chip provides high-resolution mapping of TFAM binding and reveals a global loss of sequence specificity following UVC exposure.(3) The combination of in vitro TFAM DNA biophysical approaches, combined with cellular responses (gene expression, mtDNA turnover), provides a coherent multi-scale view.(4) The authors demonstrate that TFAM-induced compaction does not protect mtDNA from UVC lesions, an important contribution given assumptions about TFAM providing protection.Weaknesses:(1) The authors show a decrease in mtDNA levels and increased lysosomal colocalization but do not define the pathway responsible for degradation. Distinguishing between replication dilution, mitophagy, or targeted degradation would strengthen the interpretation

We thank the reviewer for their careful reading of our manuscript and thoughtful suggestions. We agree that distinguishing between replication dilution, mitophagy, and/or targeted degradation would strengthen our understanding of how UV-induced DNA damage is handled in the mitochondria. Currently we are undertaking experiments to tease this apart, but consider the scope of those experiments to be beyond this manuscript and expect to publish them in a subsequent paper rather than this one. We added text explicitly stating that these possibilities are not distinguished by our results in pages 8-9 in the Discussion under the subsection ‘Mitochondria respond to UVC-induced mtDNA damage in the absence of apparent mitochondrial dysfunction’.

(2) The sudden induction of mtDNA replication genes and transcription at 24 h suggests that intermediate timepoints (e.g., 12 hours) could clarify the kinetics of the response and avoid the impression that the sampling coincidentally captured the peak.

We agree and have added additional timepoints of 12 hours and 18 hours post exposure. We have updated Figure 2 to include the new data and have added text on page 4 to include these results.

(3) The authors report no loss of mitochondrial membrane potential, but this single measure is limited. Complementary assays such as Seahorse analysis, ATP quantification, or reactive oxygen species measurement could more fully assess functional integrity.

We focused on membrane potential because loss of membrane potential is such a well-understood of mechanism for triggering mitophagy, but agree that these additional measurements are useful. We have added experiments to assess ATP levels, but did not see changes; we have added this data to Figure 2. We have also added text highlighting that we previously assessed mtROS following the same levels of UV exposure and observed no changes (in the results section on page 5 and in the discussion section on page 9). Given that we observe no changes in membrane potential or ATP, we have opted to not move forward with Seahorse analysis for the purposes of this paper.

(4) The manuscript briefly notes enrichment of TFAM at certain regions of the mitochondrial genome but provides little interpretation of why these regions are favored. Discussion of whether high-occupancy sites correspond to regulatory or structural elements would add valuable context.

We agree a discussion of these findings provides context and insight into where the field is currently in understanding TFAM sequence specificity. We have updated text in the discussion (pages 9-10) to include our thoughts on the drivers of TFAM sequence specificity with regard to the discrepancy with the anisotropy data and the lack of overlap with regulatory/structural elements.

(5) It remains unclear whether the altered DNA topology promotes TFAM compaction or vice versa. Addressing this directionality, perhaps by including UVC-only controls for plasmid conformation, would help disentangle these effects if UVC is causing compaction alone.

We have added an additional control making this comparison and updated the text on page 7 in the results section. UVC by itself (without TFAM being present) does not alter the plasmid compaction; see new supplemental Figure S16.

(6) The authors provide a discrepancy between the anisotropy and binding array results. The reason for this is not clear, and one wonders if an orthogonal approach for the binding experiments would elucidate this difference (minor point).

The discrepancy between anisotropy and the binding array results is certainly unusual and contrary to previous studies that have used these arrays. In addition to the anisotropy experiments, we selected a ‘high occupancy’ and ‘low occupancy’ sequence from the binding array and performed oligomerization experiments using atomic force microscopy, which allowed us to detect small changes in cooperativity (see supplemental Figure S15). We previously only discussed this briefly in the results section on page 6, but we have now updated the discussion section (pages 9-10) to highlight this finding and put forth ideas for the field as to why we think this might be the case. While we do see that the binding array data aligns with oligomerization and cooperativity of TFAM, we still do not know what it is about these sequences that would drive such differences in TFAM binding, but we speculate that it could have something to do with flexibility of the DNA sequences.

Assessment of conclusions:The manuscript successfully meets its primary goal of testing whether TFAM protects mtDNA from UVC damage and the impact this has on the mtDNA. While their data points to an intriguing model that TFAM acts as a sensor of damaged mtDNA, the validation of this model requires further investigation to make the model more convincing. This is likely warranted for a follow-up study. Also, the biological impact of this compaction, such as altering transcription levels, is not clear in this study.

We have updated wording in the Abstract, Introduction, and elsewhere in the text (as detailed in other portions of our response) to make as explicit and clear as possible which results are supported by the in vitro versus in vivo data, and which parts are conclusions supported by the data versus hypothesized models to be tested in future work.

Impact and utility of the methods:This work advances our understanding of how mitochondria manage UVC genome damage and proposes a structural mechanism for damage "sensing" independent of canonical repair. The methodology, including the custom TFAM DNA chip, will be broadly useful to the scientific community.Context:The study supports a model in which mitochondrial genome integrity is maintained not only by repair factors, but also by selective sequestration or removal of damaged genomes. The demonstration that TFAM compaction correlates with damage rather than protection reframes an interesting role in mtDNA quality control.
**Reviewer #2 (Public review):**
Summary:King et al. present several sets of experiments aimed to address the potential impact of UV irradiation on human mitochondrial DNA as well as the possible role of mitochondrial TFAM protein in handling UV-irradiated mitochondrial genomes. The carefully worded conclusion derived from the results of experiments performed with human HeLa cells, in vitro small plasmid DNA, with PCR-generated human mitochondrial DNA, and with UV-irradiated small oligonucleotides is presented in the title of the manuscript: "UV irradiation alters TFAM binding to mitochondrial DNA". The authors also interpret results of somewhat unconnected experimental approaches to speculate that "TFAM is a potential DNA damage sensing protein in that it promotes UVC-dependent conformational changes in the [mitochondrial] nucleoids, making them more compact." They further propose that such a proposed compaction triggers the removal of UV-damaged mitochondrial genomes as well as facilitates replication of undamaged mitochondrial genomes.Strengths:(1) The authors presented convincing evidence that a very high dose (1500 J/m2) of UVC applied to oligonucleotides covering the entire mitochondrial DNA genome alleviates sequence specificity of TFAM binding (Figure 3). This high dose was sufficient to cause UV lesions in a large fraction of individual oligonucleotides. The method was developed in the lab of one of the corresponding authors (reference 74) and is technically well-refined. This result can be published as is or in combination with other data.(2) The manuscript also presents AFM evidence (Figure 4) that TFAM, which was long known to facilitate compaction of the mitochondrial genome (Alam et al., 2003; PMID 12626705 and follow-up citations), causes in vitro compaction of a small pUC19 plasmid and that approximately 3 UVC lesions per plasmid molecule result in a slight, albeit detectable, increase in TFAM compaction of the plasmid. Both results can be discussed in line with a possible extrapolation to in vivo phenomena, but such a discussion should include a clear statement that no in vivo support was provided within the set of experiments presented in the manuscript.

We thank this reviewer for their careful reading and interpretation of the manuscript. We agree that discussion of in vivo implications and extrapolations need clear statements indicating where there is not currently in vivo support. We have updated the text throughout the paper to include this.

Weaknesses:Besides the experiments presented in Figures 3 and 4, other results do not either support or contradict the speculation that TFAM can play a protective role, eliminating mitochondrial genomes with bulky lesions by way of excessive compaction and removing damaged genomes from the in vivo pool.To specify these weaknesses:(1) Figure 1 - presents evidence that UVC causes a reduction in the number of mitochondrial spots in cells. The role of TFAM is not assessed.

We are working to understand the role of TFAM in vivo following UV irradiation, but believe that work should be included in follow up studies rather than this publication.

(2) Figure 2 - presents evidence that UVC causes lesions in mitochondrial genomes in vivo, detectable by qPCR. No direct assessment of TFAM roles in damage repair or mitochondrial DNA turnover is assessed despite the statements in the title of Figure 2 or in associated text. Approximately 2-fold change in gene expression of TFAM and of the three other genes does not provide any reasonable support to suggestion about increased mitochondrial DNA turnover over multiple explanations on related to mitochondrial DNA maintenance.

We agree and have updated the title of Figure 2 to better reflect the findings outlined in the figure as well as the text.

The new title is, “UVC causes mtDNA damage that decreases over time and is associated with upregulation of mtDNA replication genes, in the absence of apparent mitochondrial dysfunction.”

We agree that there are numerous mechanistic hypotheses that could explain the decrease in mtDNA damage over time. In Figure 1, we show that there is an overall decrease in mtDNA spots, and an increase in mtDNA-lysosome colocalization, suggestive of mtDNA degradation, which could serve to remove damaged genomes. One possibility is that TFAM is playing a role in the damage removal (but not repair per cell as these lesions are not repaired). Another is changes in mtDNA turnover via increasing the replication machinery in order the synthesize non-damaged mtDNA molecules to dilute out damage. These and other possibilities are not mutually exclusive. We have added text (pages 8-9) to make explicit that additional work will be required to distinguish these possibilities. We note that we have also added an additional experiment showing that TFAM knockdown affects mtDNA damage at baseline, as well as after UVC exposure (Figure 5J).

(3) Figure 5. Shows that TFAM does not protect either mitochondrial nucleoids formed in vitro or mitochondrial DNA in vivo from UVC lesions as well as has no effect on in vivo repair of UV lesions.

We agree that Figure 5 shows that TFAM does not protect DNA from UVC-induced lesions, and that a roughly 2-fold increase in TFAM protein does not alter damage reduction over time. We have added new data showing that in vivo, knockdown of TFAM results in an increase in baseline (control conditions) mtDNA damage, and also alters the rate of decrease of mtDNA damage over time after UVC (Figure 5J).

(4) Figure 6: Based on the above analysis, the model of the role of TFAM in sensing mtDNA damage and elimination of damaged genomes in vivo appears unsupported.

We have updated the legend for Figure 6 in which we outline our hypothesized role of TFAM in sensing mtDNA damage to ensure that readers know this has yet to be fully tested in vivo. We have also updated the Figure legend title from “proposed model” to “hypothesized model,” and changed the wording in the conclusion section (page 11) to highlight more clearly that this is a working model.

(5) Additional concern about Figure 3 and relevant discussion: It is not clear if more uniform TFAM binding to UV irradiated oligonucleotides with varying sequence as compared to non-irradiated oligonucleotides can be explained by just overall reduced binding eliminating sequence specific peaks.

We do not believe this is the case given the similar K_D_ values for the sequences tested. In our hands and in other publications (reviewed in PMID: 34440420), it has been well established that TFAM binds damaged DNA very well—essentially just as well as nondamaged DNA or better.

Additionally, a reduction in overall binding on these DNA arrays tends to make sequence specific peaks more apparent. We ran our experiments at both 30 nM and 300 nM TFAM specifically to be able to assess this question. The 300 nM data can be found in supplemental Figure S7. In this figure, we notice that the peaks appear more uniform at the high concentration (comparing Figure 3A to Figure S7A). That is presumably because there is so much more binding happening across the array that the peaks associated with the strongest binders become less pronounced. For the sake of brevity, we have not added this reasoning to the text, but are willing to do so if the Reviewers and Editor feel that it is important to include.

**Reviewer #3 (Public review):**
Summary:The study is grounded in the observations that mitochondrial DNA (mtDNA) exhibits a degree of resistance to mutagenesis under genotoxic stress. The manuscript focuses on the effects of UVC-induced DNA damage on TFAM-DNA binding in vitro and in cells. The authors demonstrate increased TFAM-DNA compaction following UVC irradiation in vitro based on high-throughput protein-DNA binding and atomic force microscopy (AFM) experiments. They did not observe a similar trend in fluorescence polarization assays. In cells, the authors found that UVC exposure upregulated TFAM, POLG, and POLRMT mRNA levels without affecting the mitochondrial membrane potential. Overexpressing TFAM in cells or varying TFAM concentration in reconstituted nucleoids did not alter the accumulation or disappearance of mtDNA damage. Based on their data, the authors proposed a plausible model that, following UVC-induced DNA damage, TFAM facilitates nucleoid compaction, which may serve to signal damage in the mitochondrial genome.Strengths:The presented data are solid, technically rigorous, and consistent with established literature findings. The experiments are well-executed, providing reliable evidence on the change of TFAM-DNA interactions following UVC irradiation. The proposed model may inspire future follow-up studies to further study the role of TFAM in sensing UVC-induced damage.Weaknesses:The manuscript could be further improved by refining specific interpretations and ensuring terminology aligns precisely with the data presented.(1) In line 322, the claim of increased "nucleoid compaction" in cells should be removed, as there is a lack of direct cellular evidence. Given that non-DNA-bound TFAM is subject to protease digestion, it is uncertain to what extent the overexpressed TFAM actually integrates into and compacts mitochondrial nucleoids in the absence of supporting immunofluorescence data.

We would like to thank this reviewer for their comments and suggestions. We feel these specific language changes have strengthened the interpretability of the text. The TFAM overexpression cells used in this experiment were given to us by Isaac et al., who demonstrated that when TFAM was overexpressed in this specific cell line, the nucleoids were indeed more compact, measured by Fiber-seq (Isaac et al., 2024; PMID: 38347148). We have removed the claim “increased compaction” from the section title, Figure 5 legend title, and from line 322 (now on page 8), and have also added an additional sentence to ensure the reader knows these cells have been shown to have presumed increased compaction by other groups.

(2) In lines 405 and 406, the authors should avoid equating TFAM overexpression with compaction in the cellular context unless the compaction is directly visualized or measured.

We have updated the text to ensure that it is clear that this was tested by other groups. We also changed the wording to “inaccessible (presumably compacted) nucleoids.” While we did not demonstrate altered compaction in our study, we think that based on the results from Isaac et al., it is likely that there was increased compaction. In addition, some readers might not have the context to make the connection between compaction and accessibility, so eliminating all reference to compaction could obscure the point.

(3) In lines 304 and 305 (and several other places throughout the manuscript), the authors use the term "removal rates". A "removal rate" requires a direct comparison of accumulated lesion levels over a time course under different conditions. Given the complexity of UV-induced DNA damage-which involves both damage formation and potential removal via multiple pathways-a more accurate term that reflects the net result of these opposing processes is "accumulated DNA damage levels." This terminology better reflects the final state measured and avoids implying a single, active 'removal' pathway without sufficient kinetic data.

We agree and have updated the language throughout the text as well as the results heading for this section.

(4) In line 357, the authors refer to the decrease in the total DNA damage level as "The removal of damaged mtDNA". The decrease may be simply due to the turnover and resynthesis of non-damaged mtDNA molecules. The term "removal" may mislead the casual reader into interpreting the effect as an active repair/removal process.

We agree and have restructured this sentence for clarity. We do believe there is some removal happening, given the increase in mtDNA colocalization in lysosomes alongside decrease of mtDNA spots in our live cell imaging. We have written it to reflect the inclusion of removal and resynthesis of nondamaged mtDNA molecules (see pages 8-9).

**Recommendations for the authors:**

**Reviewing Editor Comments:**
The reviewers appreciate the quality of the presented data but concur that they do not support the primary claims in the title and abstract. The reviewers also realize that in vivo evidence for the model would require extensive new experimentation that goes beyond a reasonable revision. The recommendation is to change the title and significantly revise text, figure titles and legends for transparency, and conclusions within results and discussion sections.

We thank the editor and all the reviewers for their feedback. We have added additional experiments, updated text throughout the entire paper to ensure our claims are supported, and revised our title. We feel that the changes we have made have indeed made the paper stronger, more transparent, and that the evidence put forth in this paper provides support for all claims made.

**Reviewer #1 (Recommendations for the authors):**
(1) Clarify mitochondrial response kinetics by adding an intermediate (e.g., 12 hrs) recovery timepoint for transcriptional analysis to resolve when TFAM and replication genes are induced.

We have added additional timepoints of 12 and 18 hours following exposure in Figure 2. These results strengthen our finding that the nuclear transcriptional program supporting mtDNA replication appears to be activated prior to the nuclear transcriptional program supporting mitochondrial transcription, in that POLG and TFAM come up before POLRMT and ND1.

(2) Strengthen functional readouts by assessing additional parameters of mitochondrial function to substantiate the claim that UVC does not impair mitochondrial performance.

We have referenced our previously-published data on mtROS and added a measurement of ATP following UVC exposure in Figure 2.

(3) Consider exploring whether mtDNA degradation occurs via mitophagy, nucleoid-phagy, or another pathway-potentially by using inhibitors or markers of these processes.

While we agree that this is an important follow up question and are currently working on experiments to address this, those experiments are outside the scope of this manuscript.

(4) Provide additional details for the high occupancy TFAM sites. Provide brief annotation or discussion of genomic regions showing strong TFAM binding under non-irradiated conditions that are lost during UVC treatment. This would be helpful to the field as a whole.

We have updated our discussion section to include this.

(5) Include or discuss a control using UVC irradiated pUC19 without TFAM to confirm that observed compaction categories are TFAM dependent rather than an UVC induced DNA distortion.

We have added in a supplemental figure (Figure S16) containing comparison of area analysis of control pUC19 and UV-irradiated pUC19 and we have added associated text in the results section of the paper.

(6) It would be interesting to explore the link between compaction to transcriptional output. In the TFAM overexpression model, the authors could measure expression of mtDNA encoded transcripts (e.g., ND1, COX1) to connect increased compaction with altered mitochondrial transcription.

While we agree that understanding how the compactional status alters mitochondrial transcription is worthwhile, we believe this is beyond the scope of this paper. Furthermore, this connection has previously been shown by Bruser et al., 2021 (PMID: 34818548) who showed that more compact nucleoids are not undergoing active transcription. It will be interesting to see in future work if mtDNA damage drives changes in both compaction as well as transcriptional activity.

(7) Clarify quantitative presentation in figure 2F to explicitly note whether the observed increase in fluorescence intensity was statistically insignificant and confirm that the assay sensitivity is sufficient to detect small potential changes. As presented it is not clear if there is a change.

We have changed the presentation of Figure 2F. There is a slight increase in membrane potential at the 24-hour time point and we have made that clear in the text as well. We included FCCP as a (standard) positive control, for which we can detect the associated decrease in membrane potential for. While it is always possible that a very small decrease occurred that we were unable to detect, we note that none of the six UVC-exposed groups that we tested even trended towards a decrease in MMP, making it less likely that there was an effect that we simply lacked the power or sensitivity to detect.

(8) It would be interesting if the authors can comment on whether TFAM induced compaction after UVC might shield mtDNA from other, repairable lesions (e.g., oxidative or alkylation damage), offering a broader context for this mechanism beyond just UVC.

In theory, we believe this is possible. It will also be interesting to see if the increased compaction following UVC also protects or shields the mtDNA from other enzymatic processes, such as repair proteins that may be searching for repairable lesions such as oxidative or alkylation damage. In this case, it seems as though the increased compaction would prevent the repair from happening at genomes harboring damage.

In this study we show with our in vitro nucleoids that the increased compaction does not protect against UVC, but this is likely because UVC does not need physical access to the DNA in order to damage it, as the wavelengths of UVC (centered in this case at 254nm) are readily absorbed by proteins and thus can go right through the proteins. Currently, we know that increased compaction by TFAM makes the DNA inaccessible to the enzymes required to methylate DNA used in Fiber-seq (PMID: 38347148), but we do not know if the compaction is tight enough to prevent ROS or alkylating agents from damaging the DNA. We have updated text in the discussion on page 10 to highlight some of these ideas.

**Reviewer #2 (Recommendations for the authors):**
Please, go over all display items and text and clarify details that can help readers to understand important specifics of the experiments. Examples are provided below:(1) Abstract and Introduction - indicate species and cell line

We have updated the text to include this information.

(2) Table 1 "TFAM KD measurements"- title and footnotes are entirely cryptic. Please, clarify the experimental design, question(s) addressed and conclusions drawn from data.

We have updated the title of Table 1 to "Binding of TFAM to array sequences, measured using fluorescence anisotropy,” and clarified the footnotes to make sure it is clear which sequences were selected for AFM oligomerization experiments.

(3) Figure 3 and Material and Methods - specify UVC dose.

We have added this information to both the figure legend and the methods section.

(4) Figure 4 - specify UVC dose.

We have added this information to the figure legend.

(5) Figure 5. Panel B indicate which band is TFAM and which is HA-tag; Indicate clearly which panel is showing in vivo or in vitro results.

We have updated the figure to label the untagged TFAM and HA-tagged TFAM and changed the panel titles to specify if they are in vivo results.